# Learning to Remember, Learn, and Forget in Attention-Based Models

Djohan Bonnet [1]  Jamie Lohoff [1 2]  Jan Finkbeiner [1 2]  Elidona Shiqerukaj [2]  Emre Neftci [1 2]

## Abstract

In-Context Learning (ICL) in transformers acts as an online associative memory and is believed to underpin their high performance on complex sequence processing tasks. However, in gated linear attention models, this memory has a fixed capacity and is prone to interference, especially for long sequences. We propose Palimpsa, a self-attention model that views ICL as a continual learning problem that must address a stability-plasticity dilemma. Palimpsa uses Bayesian metaplasticity, where the plasticity of each attention state is tied to an importance state grounded by a prior distribution that captures accumulated knowledge. We demonstrate that various gated linear attention models emerge as specific architecture choices and posterior approximations, and that Mamba2 is a special case of Palimpsa where forgetting dominates. This theoretical link enables the transformation of any non-metaplastic model into a metaplastic one, significantly expanding its memory capacity. Our experiments show that Palimpsa consistently outperforms baselines on the Multi-Query Associative Recall (MQAR) benchmark and on Commonsense Reasoning tasks.

## 1. Introduction

Transformers are the workhorse of generative AI, underpinning breakthroughs in large language modeling (Brown et al., 2020; Chowdhery et al., 2023), computer vision (Dosovitskiy et al., 2021), robotics (Brohan et al., 2022), and scientific domains like chemistry (Schwaller et al., 2019) and biology (Jumper et al., 2021). Central to this success is the *self-attention* mechanism (Vaswani et al., 2017), which allows a model to capture interactions among all tokens in a sequence in a highly parallelizable manner. However, this mechanism requires caching key-value (KV) pairs for each input token, causing memory and computational costs to grow linearly and quadratically, respectively, with sequence length. Consequently, long-context data scenarios pose strong technical challenges for classical Transformer architectures, especially at the edge (Kim et al., 2023).

A promising direction to simultaneously address the computational and memory bottlenecks is to adopt *fixed-size* attention memories, such as in linear transformers (Schlag et al., 2021) and state space models (Gu & Dao, 2023). These trade the dynamically growing KV cache with a fixed-size memory state that is updated at each token. A key insight is their ability to perform *in-context learning (ICL)* (Brown et al., 2020), whereby the model effectively "learns" at test time by processing contextual examples within the input. ICL essentially tackles a continual and online learning problem, not unlike how the brain dynamically adjusts its weights through neural and synaptic plasticity. In this view, learning with fixed-size memory without revisiting past states and inputs (*i.e.* replay) can lead to catastrophic interference (McCloskey & Cohen, 1989) and further deteriorate with sequence length. This interference contributes to the limited capacity of linear transformers (Schlag et al., 2021).

The concept of *metaplasticity* from neuroscience (Abraham & Bear, 1996) posits that the degree of plasticity is itself adaptive to preserve prior knowledge. This inspired practical algorithms to solve catastrophic forgetting by dynamically adjusting plasticity (Zenke et al., 2017; Kirkpatrick et al., 2017b; Benna & Fusi, 2016). In this work, we pose ICL as a continual learning problem that can be solved through metaplasticity, building on prior work that associated ICL and gradient descent (Akyürek et al., 2023; Von Oswald et al., 2023b). However, existing metaplasticity methods are unsuitable for transformers because they rely either on clearly demarcated tasks and associated labels, growing memory, or lack differentiability (Kudithipudi et al., 2023). As we aim to introduce metaplastic methods embedded in transformer architectures, differentiability is necessary for training. One natural solution to these challenges is to formulate memory updates as a process of *Bayesian Gradient Descent (BGD)*, ensuring that each update balances prior knowledge with new evidence (Zeno et al., 2021). BGD adjusts parameters according to their uncertainty in an online

---

[1]Forschungszentrum Jülich , Germany [2]RWTH Aachen, Germany. Correspondence to: Djohan Bonnet <d.bonnet@fz-juelich.de>, Emre Neftci <e.neftci@fz-juelich.de>.

*Proceedings of the 43rd International Conference on Machine Learning*, Seoul, South Korea. PMLR 306, 2026. Copyright 2026 by the author(s).

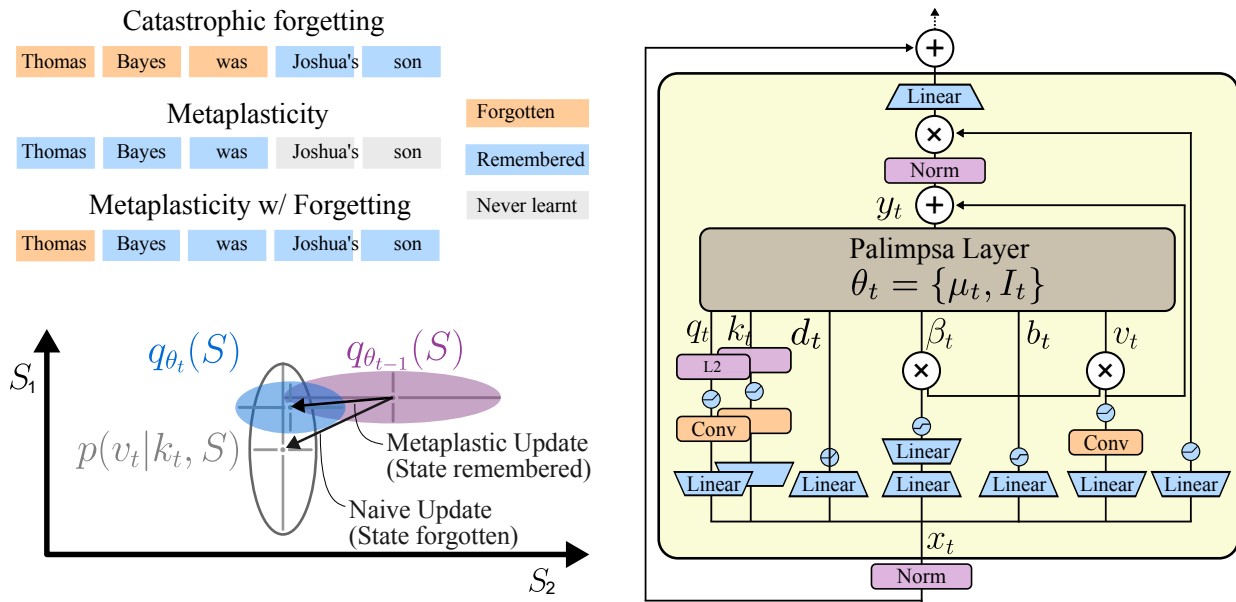

*Figure 1.* **Bayesian Metaplasticity Attention.** Self-attention in autoregressive transformers is inherently a continual learning problem, and as such can suffer from catastrophic forgetting. Metaplasticity dynamically modifies the learning rate to preserve important prior information. (Bottom-left) Illustration of Bayesian metalearning: $q_{\theta_t}$ is the (variational) distribution over memory states $S$ at time step $t$. (Right) We derive Palimpsa, a new attention block based on an online Bayesian posterior, preventing both catastrophic forgetting and remembering using metaplasticity.

fashion. However, BGD and related metaplasticity methods suffer from catastrophic *remembering* (Kaushik et al., 2021), induced by a loss of plasticity that *suppresses* the ability to learn new knowledge to preserve old ones. For ICL, this would imply that the model becomes unable to incorporate new information past a critical sequence length, which is often cited as a key limitation of state space models (Behrouz et al., 2024). The recent Metaplasticity from Synaptic Uncertainty (MESU) solves this problem by leveraging a Bayesian framework to also enable *forgetting* by discarding stale or unused information (Bonnet et al., 2025). The degree of this *palimpsest* property [1] can be adjusted through a Bayesian prior, which effectively controls the time horizon of the memory.

Building on MESU, we introduce *Palimpsa*, a dual-state attention block that performs metaplastic Bayesian updates to a fixed-size attention memory. Palimpsa mitigates in-context catastrophic forgetting by adjusting per-state update magnitude based on memory uncertainty, preserving critical past information. By releasing information predicted as stale, it also prevents in-context catastrophic remembering. The key to deriving Palimpsa is the formulation of self-attention as an optimization of an inner variational free energy at test-time, for which the variational posterior updates can be

---

[1] A palimpsest is a writing surface where the original text is scraped or washed off to be reused, especially used in ancient works when parchment was of limited supply – similarly to the fixed-size memory in Palimpsa

analytically computed. With our custom-developed kernels, Palimpsa scales efficiently on GPUs with speeds comparable to Mamba1 (Gu & Dao, 2023).

Furthermore, we show that existing gated linear models can be derived from specific assumptions on the variational posterior, unifying several prior works within a common mathematical framework. Specifically, we demonstrate a continuum between Mamba2 and Palimpsa, suggesting a hybrid pipeline: large-scale pre-training for efficiency, followed by metaplastic fine-tuning with Palimpsa to expand its memory capacity. We validate our approach on synthetic, commonsense reasoning, and long context benchmarks using two backbones: Palimpsa-D (Deltanet-based) and Palimpsa-M (Mamba2-based). Our results demonstrate that metaplasticity consistently improves performance over non-metaplastic baselines across various memory sizes.

Our specific contributions are:

- The introduction of the concept of metaplasticity in ICL to mitigate in-context catastrophic forgetting.
- Palimpsa, a model that prevents in-context catastrophic remembering by gradually releasing outdated knowledge, with scalable custom kernels.
- A mathematical framework that casts several gated linear attention and state space models as special cases of a common Bayesian framework.
- Methods to transform non-metaplastic models into metaplastic ones.

## 1.1. Related Work

**Continual Learning Methods in Neural Sequence Models** Continual Learning (CL) methods can be broadly categorized into complementary mechanisms for CL systems (McClelland et al., 1995) namely *replay-based* ((Buzzega et al., 2020), (Lopez-Paz & Ranzato, 2017)), *dynamic network expansion* ((Rusu et al., 2016), (Wang et al., 2024)), and *regularization-based* methods ((Zenke et al., 2017), (Kirkpatrick et al., 2017a)). CL has been previously considered in State Space Models (SSM) (Cheng et al., 2024; Zhang et al., 2024), but did not study the ICL aspects of this problem. Other work applied modern transformer and SSM models to improve on CL benchmarks (Thengane et al., 2022). To provide a local and efficient method to solve the stability-plasticity dilemma using finite size memory in ICL, we focus specifically on regularization-based CL methods. This allows us to tackle general language modeling tasks without an explicitly constructed CL structure, namely language.

A significant body of related work interprets In-Context Learning (ICL) as Bayesian inference (Xie et al., 2021; Wies et al., 2023; Arora et al., 2024a; Hahn & Goyal, 2023; Zhang et al., 2023; Falck et al., 2024). These studies generally posit that the Transformer architecture implicitly approximates a Bayesian learner in context at the system level, effectively performing inference over latent tasks or concepts. Our approach differs fundamentally, as we explicitly derive a new gated linear attention mechanism directly from Bayesian principles. Bayesian learning and metaplasticity are closely linked in the hypothesis that synapses maintain uncertainty estimates (Aitchison et al., 2021) to dynamically adjust their plasticity. This perspective inspired metaplastic CL methods like MESU (Bonnet et al., 2025). This type of metaplasticity can be written in a local and linear fashion. In Palimpsa, we take advantage of these properties to allow scalable training on GPUs using the techniques employed for gated state-space models (Gu & Dao, 2023).

**Gated Linear Recurrence Models and Metaplasticity** This type of metaplasticity is strikingly related to existing gated linear attention models and ICL. (Akyürek et al., 2023; Von Oswald et al., 2023a) demonstrated that the forward pass through the linear attention mechanism can be framed as an iterative gradient descent on a local regression loss function, thus enabling test-time adaptation of the model underpinning ICL. This showed how the choice of the loss function directly influenced the resulting gating and forgetting mechanisms (Yang et al., 2024). The perspective of gradient descent local loss functions was further elaborated in Mesa-optimization and Mesanet (Von Oswald et al., 2023b; 2025), Test-Time Training (Sun et al., 2023), Gated Deltanets (Yang et al., 2025), Longhorn (Liu et al., 2025) and Titans (Behrouz et al., 2024).

Palimpsa shares similarity with Longhorn (Liu et al., 2025) and Mesanet (Von Oswald et al., 2025). Longhorn (Liu et al., 2025) explicitly leverages this online learning relationship to derive its gating mechanisms. While input gating enables the model to control which information is compressed into the memory matrix achieving a form of suboptimal stability-plasticity trade off, it lacks an efficient mechanism to protect important information.

Specifically, Longhorn introduces a trainable, input-dependent importance factor that weights the loss function (which we later refer to as $\beta_t$) and dictates the stability-plasticity ratio for each row of the fixed-size memory matrix, effectively freezing a row for a given input when a value is zero. In Longhorn, the plasticity rate remains constant for every memory state within a sequence, meaning there is no in-context metaplasticity.

In contrast, our Bayesian metaplasticity perspective dictates a plasticity rate that changes *in-context*. This is achieved using a second state, as in the Mesanet. Mesanet (Von Oswald et al., 2025) seeks the optimal solution of an in-context regression objective, leading to an update that is strikingly similar to Palimpsa. This is because Bayesian and frequentist solutions to linear regression problems lead to identical estimates of the mean. However, Mesanet addresses a simplified objective where the stability-plasticity ratio is fixed for each memory row. Consequently, every state in a row shares the same learnable plasticity. This prevents true metaplasticity, which requires dynamically adapting the learning rate of every individual state. This structural simplification enables Mesanet to focus on capturing synaptic correlations by computing the inverse covariance matrix using a parallelized, iterative gradient descent. However, this numerical approximation comes at the computational cost of iterative inner-loop updates. To maintain throughput and the spirit of metaplasticity, where synaptic learning rates are independent, Palimpsa relies on a diagonal approximation of the covariance matrix. By using the resulting variance as a learning rate, Palimpsa closely aligns with the neuroscience model proposed by (Aitchison et al., 2021).

Finally, because Palimpsa is inherently Bayesian, the attention head provides output uncertainty that could be leveraged in future work to improve overall model performance.

## 2. Background and Methods

### 2.1. Background: Self-Attention Heads and Metaplastic Memory

The decoder self-attention (Vaswani et al., 2017) autoregressively maps the sequence $\{\boldsymbol{x}_t\}_{t=1}^L$ into $\{\boldsymbol{y}_t\}_{t=1}^L$. Each token $\boldsymbol{x}_t$ is projected into key, query, and value vectors $\boldsymbol{k}_t, \boldsymbol{q}_t \in \mathbb{R}^{d_k}, \boldsymbol{v}_t \in \mathbb{R}^{d_v}$. Self-attention computes a weighted combination of the values $\boldsymbol{V}_t = [\boldsymbol{V}_{t-1}, \boldsymbol{v}_t]$ based on the similarity

between each query $\boldsymbol{q}_t$ and the keys $\boldsymbol{K}_t = [\boldsymbol{K}_{t-1}, \boldsymbol{k}_t]$. Concretely, this can be expressed as:

$$\boldsymbol{y}_t = \boldsymbol{V}_t \, \text{Softmax}\big(\boldsymbol{K}_t^\top \boldsymbol{q}_t\big).$$

The similarity between queries and keys (the "look-up" step) determines the extent to which each value contributes to the output. Linear Transformers (Schlag et al., 2021) omit the softmax, in which case the KV cache can be replaced by a fixed-size attention matrix, updated in-place for every new incoming token with a "Hebbian" update:

$$\boldsymbol{S}_t = \boldsymbol{S}_{t-1} + \boldsymbol{v}_t \otimes \boldsymbol{k}_t, \text{ and } \boldsymbol{y}_t = \boldsymbol{S}_t \, \boldsymbol{q}_t, \text{ where } \boldsymbol{S}_t \in \mathbb{R}^{d_v \times d_k}.$$

However, if the new key $\boldsymbol{k}_t$ is not orthogonal to previously stored keys, the update will partially overwrite older memories. Schlag et al. demonstrated the limited capacity of linear transformers and implemented an error-correcting delta rule to mitigate this issue. This idea has been later expanded to improve performance and trainability (Sun et al., 2024; Liu et al., 2025; Yang et al., 2025). Given this interference, should the model prioritize learning to map new key–value pairs, or preserve the relationships from earlier pairs?

Recently, Bonnet et al. proposed MESU, a mathematically grounded approach to solve such dilemma from a Bayesian framework. MESU formalizes forgetting to prevent a vanishing learning rate thus avoiding catastrophic remembering (Kaushik et al., 2021). Building on this idea, we treat the self-attention memory update as a Bayesian inference problem that includes formalized forgetting.

## 2.2. Derivation of Palimpsa

**Bayesian Formulation of the Attention Objective**  For a given time step, the attention head can be defined as $p(\boldsymbol{v}|\boldsymbol{k}, \boldsymbol{\beta}, \boldsymbol{S})$. Given the inputs $\boldsymbol{k} \in \mathbb{R}^{d_k}$ and importance factor $\boldsymbol{\beta} \in \mathbb{R}^{d_v}$, the attention head models the distribution of values $\boldsymbol{v} \in \mathbb{R}^{d_v}$ conditioned on the state $\boldsymbol{S} \in \mathbb{R}^{d_v \times d_k}$. Using Gaussian assumptions, the objective function (linear regression) of the attention head is defined as the negative log-likelihood:

$$-\log p(\boldsymbol{v} \mid \boldsymbol{k}, \boldsymbol{\beta}, \boldsymbol{S}) = \tfrac{1}{2}\|\boldsymbol{S}\boldsymbol{k} - \boldsymbol{v}\|^2_{\text{diag}(\boldsymbol{\beta})}.$$

Given $t$ tokens $\{\boldsymbol{x}_j\}_{j=1}^t$ with $\boldsymbol{x}_j \in \mathbb{R}^{d \times 1}$, we obtain $t$ data points $\{\boldsymbol{d}_j\}_{j=1}^t = \{(\boldsymbol{k}_j, \boldsymbol{\beta}_j), \boldsymbol{v}_j\}_{j=1}^t$ to minimize the above loss "in-context". Applying Bayes' rule gives the posterior distribution on $\boldsymbol{S}$, which can be expressed in the recursive form:

$$p(\boldsymbol{S}|\boldsymbol{d}_{1:t}) = \frac{p(\boldsymbol{d}_t|\boldsymbol{S}) \cdot p(\boldsymbol{S}|\boldsymbol{d}_{1:t-1})}{p(\boldsymbol{d}_t|\boldsymbol{d}_{1:t-1})}. \quad (1)$$

Since $\boldsymbol{q}_t$ is independent of $\boldsymbol{S}$, the output $\boldsymbol{y}_t$ can be defined as:

$$\boldsymbol{y}_t = \mathbb{E}_{p(\boldsymbol{S}|\boldsymbol{d}_{1:t})}\left[\boldsymbol{S}\boldsymbol{q}_t\right] = \mathbb{E}_{p(\boldsymbol{S}|\boldsymbol{d}_{1:t})}\left[\boldsymbol{S}\right]\boldsymbol{q}_t = \boldsymbol{\mu}_t \boldsymbol{q}_t. \quad (2)$$

Catastrophic remembering occurs when the prior $p(\boldsymbol{S}|\boldsymbol{d}_{1:t-1})$ becomes overly concentrated as $t$ increases, causing the likelihood $p(\boldsymbol{d}_t|\boldsymbol{S})$ to have a negligible impact on the posterior, inhibiting the integration of new information. This can be prevented by truncating the posterior to the last $N$ tokens:

$$p(\boldsymbol{S}|\boldsymbol{d}_{t-N+1:t}) \propto p(\boldsymbol{d}_t|\boldsymbol{S})\underbrace{p(\boldsymbol{S}|\boldsymbol{d}_{t-N:t-1})}_{\text{Learning}} \cdot \underbrace{\frac{1}{p(\boldsymbol{d}_{t-N}|\boldsymbol{S})}}_{\text{Forgetting}}$$

where $N$ now acts as a memory window size (Bonnet et al., 2025). In an online scenario, $p(\boldsymbol{d}_{t-N}|\boldsymbol{S})$ is unavailable at time $t$. Thus, we approximate this truncation by the weighted posterior (See Appendix A.2):

$$p_w(\boldsymbol{S}|\boldsymbol{d}_{1:t}) \propto p(\boldsymbol{d}_t|\boldsymbol{S})p_w(\boldsymbol{S}|\boldsymbol{d}_{1:t-1}) \cdot \left(\frac{p_w(\boldsymbol{S}|\boldsymbol{d}_{1:t-1})}{p(\boldsymbol{S})}\right)^{-\frac{1}{N}}$$

We refer to $p_w$ as the weighted posterior because the influence of historical data decays over time. This suggests a simple interpretation: at each time step, we discard a fraction $\frac{1}{N}$ of the accumulated prior weight. Effectively, the weighted posterior never carries more weight than that of a truncated posterior with a memory window of size $N$. Thus, with a context window $N$ that reflects the maximum memory capacity of the attention head, catastrophic remembering is avoided. Inspired by common practice in state space models, we introduce input-dependence to this memory window, denoting it $N_t$. Our Bayesian formulation naturally defines the forgetting gate as $\alpha_t := \left(1 - \frac{1}{N_t}\right)$.

To align with the parameterization of current state-of-the-art architectures (Gu & Dao, 2023), we express this gate as $\alpha_t = \exp(-Ad_t)$, where $d_t \in \mathbb{R}_+$ represents the input-dependent step size and $A \in \mathbb{R}_+$ is a learnable parameter, ensuring $\alpha_t \in (0, 1]$. This explicit mapping provides a Bayesian interpretation of the effective dependency range of the attention head.

**Bayesian Attention as an Optimization Problem**  While the weighted posterior $p_w(\boldsymbol{S}|\boldsymbol{d}_{1:t})$ is analytically tractable in our setting, we deliberately employ Variational Inference (VI) to cast the update as an optimization problem (Blundell et al., 2015). Crucially, in this conjugate setting, VI recovers the exact solution. We adopt this formulation to demonstrate that various gated linear attention models emerge as specific choices of architecture and posterior approximations. For each time step $t$, we define $q_{\boldsymbol{\theta}_{t,i}}(\boldsymbol{S}_i)$, $i = 1, \ldots, d_v$, a variational distribution over $\boldsymbol{S}_i$ parameterized by $\boldsymbol{\theta}_{t,i}$. We minimize the Kullback–Leibler (KL) divergence between $q_{\boldsymbol{\theta}_t}$ and the true posterior by optimizing $\boldsymbol{\theta}_{t,i}$:

$$\boldsymbol{\theta}_{t,i} = \underset{\boldsymbol{\theta}}{\arg\min} \, D_{\text{KL}}\left[q_{\boldsymbol{\theta}}(\boldsymbol{S}_i) \,\|\, p_w(\boldsymbol{S}_i|\boldsymbol{d}_{1:t})\right], \, i = 1, \ldots, d_v.$$

Here, $q_{\boldsymbol{\theta}_{t,i}}$ is modeled as a multivariate Gaussian of dimension $d_k$:

$$q_{\boldsymbol{\theta}_{t,i}}(\boldsymbol{S}) \sim \mathcal{N}(\boldsymbol{\mu}_{t,i}, \boldsymbol{\Sigma}_{t,i}), \text{ where } \boldsymbol{\theta}_{t,i} = \{\boldsymbol{\mu}_{t,i}, \boldsymbol{\Sigma}_{t,i}\},$$

$$\boldsymbol{\mu}_{t,i} \in \mathbb{R}^{d_k}, \quad \boldsymbol{\Sigma}_{t,i} \in \mathbb{R}^{d_k \times d_k}, \quad i = 1, \dots, d_v.$$

Using Bayes' theorem and the definition of KL divergence, finding the optimal $\boldsymbol{\theta}_{t,i}$ is equivalent to minimizing the free energy:

$$\mathcal{F}_{t,i} = D_{\mathrm{KL}}\left[q_{\boldsymbol{\theta}_{t,i}}(\boldsymbol{S}_i) \,\|\, p(\boldsymbol{S}_i)\right] - \mathbb{E}_{q_{\boldsymbol{\theta}_{t,i}}(\boldsymbol{S}_i)}[\sum_{s=1}^{t} \log p(\boldsymbol{d}_s|\boldsymbol{S}_i)].$$

With Gaussian assumptions, $\mathcal{F}_{t,i}$ decomposes into three components, and a term $C_{\boldsymbol{\Sigma}}$ that depends solely on the covariance:

$$\mathcal{F}_{t,i}(\boldsymbol{\mu}_i) = \underbrace{\frac{\beta_{t,i}}{2}\|\boldsymbol{\mu}_i^T \boldsymbol{k}_t - v_{t,i}\|^2}_{\text{plasticity}} + \underbrace{(1-\alpha_t)\frac{\|\boldsymbol{\mu}_i\|^2}{2\sigma_{prior}^2}}_{\text{forgetting}}$$
$$+ \underbrace{\tfrac{1}{2}\alpha_t(\boldsymbol{\mu}_{t-1,i} - \boldsymbol{\mu}_i)^T \boldsymbol{\Sigma}_{t-1,i}^{-1}(\boldsymbol{\mu}_{t-1,i} - \boldsymbol{\mu}_i)}_{\text{stability}} + C_{\boldsymbol{\Sigma}}. \quad (3)$$

where $\alpha_t := (1 - \frac{1}{N_t})$. The first term contributes to adding new knowledge and is similar to other fixed-term memories. The second introduces the learning window. Through its dependence on $N$, it determines how far in the past the memories are stored, and then forgotten. The third determines the plasticity rate based on the importance of the synapse. $\mathcal{F}_{t,i}$ is analytically tractable, and its minimum with respect to $\boldsymbol{\mu}_{t,i}$ and $\boldsymbol{\Sigma}_{t,i}$ can be computed in closed form by setting its gradient to zero without requiring any approximation (see Appendix A.3).

**Palimpsa Layer and Architecture** To derive Palimpsa we use the reparameterization trick: $\boldsymbol{S}_i = \boldsymbol{\mu}_i + \boldsymbol{A}_i\boldsymbol{\epsilon}_i$, with $\boldsymbol{A}_i\boldsymbol{A}_i^\top = \boldsymbol{\Sigma}_{t,i}$, where $\boldsymbol{\epsilon}_i \sim \mathcal{N}(0, \mathbb{I})$. We find the optimal parameters $\boldsymbol{\mu}_{t,i}$ and $\boldsymbol{\Sigma}_{t,i}$, by setting the corresponding gradients to zero:

$$\frac{\partial \mathcal{F}_{t,i}}{\partial \boldsymbol{\mu}_{t,i}} = \vec{0}, \quad \text{and} \quad \frac{\partial \mathcal{F}_{t,i}}{\partial \boldsymbol{A}_{t,i}} = \boldsymbol{0}. \quad (4)$$

For computational tractability, Palimpsa only keeps the diagonal term of the covariance matrices ($\boldsymbol{\Sigma}_{t,i} = \mathrm{diag}(\boldsymbol{\sigma}_{t,i}^2)$). Under these conditions, the solution to Eqs. (4) results in the following **Palimpsa update equation**:

$$\boldsymbol{I}_t = \alpha_t \boldsymbol{I}_{t-1} + (1-\alpha_t)\boldsymbol{I}_{prior} + \beta_t \otimes \boldsymbol{k}_t^2$$
$$\boldsymbol{\mu}_t = \alpha_t \frac{\boldsymbol{I}_{t-1}}{\boldsymbol{I}_t} \odot \boldsymbol{\mu}_{t-1} + \frac{1}{\boldsymbol{I}_t} \odot \left[(\beta_t \odot \boldsymbol{v}_t) \otimes \boldsymbol{k}_t\right], \quad (5)$$

where $\boldsymbol{I}_t := \frac{1}{\boldsymbol{\sigma}_t^2}$ a precision matrix representing the importance of each state, and $I_{prior}$ is the importance prior. The full derivation can be found in Appendices A.3–A.5. This update rule is scalably computed chunk-wise using an associative scan (see Appendix A.12). Our experiments utilize

two Palimpsa variants. The first, Palimpsa-D, serves as a drop-in replacement for the Gated Delta Rule (Yang et al., 2025), differing only by an additional projection for $\beta_t$ and an intra-layer residual term. The second, Palimpsa-M, builds on Mamba2 (Dao & Gu, 2024), where the difference lies in the definition of $\beta_t$. Consequently, the metaplasticity ablation Palimpsa-M (w/o Meta) is equivalent to Mamba2 (see also next section). A crucial distinction in Palimpsa-M is that the term $d_t$, used to define the forgetting gate $\alpha_t = e^{-Ad_t}$, also functions as an input gate; this implies that new information cannot be integrated without forgetting. Further architectural details are provided in Appendix A.1.

## 2.3. Bayesian Inference at Test-Time as a General Framework of Gated Models

Here, we revisit prior gated models in the light of the Bayesian view of ICL.

A major challenge in minimizing $\mathcal{F}_{t,i}$ is calculating the inverse of the matrix $\boldsymbol{\Sigma}_{t-1,i}^{-1} \in \mathbb{R}^{d_k \times d_k}$. Von Oswald et al. use the Sherman-Morrison formula to iteratively compute it, but this approach doesn't scale well. Another approach used in Mesanet is to use conjugate gradients (Von Oswald et al., 2025). However, both methods require $\beta_t \in \mathbb{R}$ (a scalar) constant for all $i$ for tractability, since a vector $\boldsymbol{\beta}_t$ would require the inversion for every row of the state matrix. While using a scalar $\beta_t$ simplifies inversion, it forces the stability term in Eq. 3 to be the same for every $i$, preventing per-parameter plasticity.

Longhorn, Deltanets and Palimpsa use at least one approximation for tractability. A common simplification is to assume the matrix $\boldsymbol{\Sigma}_{t-1,i}^{-1}$ is diagonal (meaning it only has values on its main diagonal). Both Longhorn and Palimpsa leverage this diagonal simplification to support a vector-valued $\boldsymbol{\beta}_t$. In contrast, Deltanets solve their objective by assuming the loss is linear around the current states. In Palimpsa, we designate this diagonal $\boldsymbol{I}_{t-1,i} \in \mathbb{R}^{d_k}$ as the "importance" of a given synapse because its inverse provides its optimal learning rate. Tab. 1 summarizes the resulting update rules derived from these specific architecture choices and posterior approximations. For all gated models but Palimpsa and Mesanet in Tab. 1, the importance is fixed, so $\boldsymbol{I}_{t-1,i} = \vec{1}$. In other words, their stability term is constant, meaning they have no metaplasticity.

This analysis also reveals that Mamba2 is a special case of Palimpsa. In the Palimpsa update (Eq. 5), when the forgetting rate is high, the posterior uncertainty remains close to the initial prior, and $\boldsymbol{I}_t \cong \boldsymbol{I}_{prior}$ is constant. The update rule then simplifies to:

$$\boldsymbol{\mu}_t = \alpha_t \boldsymbol{\mu}_{t-1} + \frac{\boldsymbol{\beta} \odot \boldsymbol{v}_t}{I_{prior}} \otimes \boldsymbol{k}_t,$$

which takes the same form as Mamba2's update rule. While Mamba2 was described as solving a negative inner-product loss (Yang et al., 2025), our Bayesian framework reveals Mamba2 as an asymptotic special case of Palimpsa, where forgetting is so strong that the importance matrix is negligible.

## 2.4. Fine-tuning Mamba2 with Metaplasticity

We exploit the above-described continuum between Mamba2 and Palimpsa to introduce the ability to upgrade any pre-trained Mamba2 model into a metaplastic Palimpsa model via fine-tuning. For 760M models Mamba2 is $> 3\times$ faster than Palimpsa. While this slowness hinders the training of large Palimpsa models from scratch, the fine-tuning ability mitigates this cost and makes full training unnecessary.

Analytically, Palimpsa matches Mamba2 when $\boldsymbol{I}_t \cong \boldsymbol{I}_{prior}$. We achieve this condition using the reparameterization $\boldsymbol{v}_t^* = \boldsymbol{\beta}_t \odot \boldsymbol{v}_t = \text{SiLU}(\boldsymbol{\theta}_v \boldsymbol{x}_t)$. By defining the Palimpsa Kernel input as $\boldsymbol{v}^*$, the $\boldsymbol{\beta}_t$ multiplication in the kernel becomes unnecessary. This allows us to fix $\boldsymbol{\beta}_t$ arbitrarily close to zero, ensuring $\boldsymbol{I}_t \cong \boldsymbol{I}_{prior}$, without forcing $\boldsymbol{\mu}_t \approx 0$. To do so we introduce a per-head learnable (but fixed in-context) $b_{scale}$ that scales the amplitudes of $\boldsymbol{\beta}_t$. For standard training, $b_{scale} = 1$; for fine-tuning, $b_{scale}$ is initialized in $[0.1, 1.0]$ so that $\boldsymbol{\beta}_t$ starts small placing the model closer to the Mamba2 limit. Intuitively, when tokens are of arbitrarily low importance but carry arbitrarily large values, Palimpsa becomes Mamba2. Thanks to this reparameterization, training can continuously deviate from Mamba2 model towards Palimpsa, as it starts learning that $\boldsymbol{\beta}_t$ should grow to exploit metaplasticity.

# 3. Experiments

## 3.1. Synthetic Experiments: MQAR

To evaluate Palimpsa's ability to manage and recall states in challenging memorization tasks prone to overwriting, we conduct experiments using the synthetic Multi Query Associative Recall (MQAR) benchmark (Arora et al., 2024b). In this task, the model is presented with pairs of key-value symbols and must retrieve the correct value associated with a specific key at test time. To evaluate the memory capacity of the model rather than its ability to select for important tokens, we selected a configuration that utilizes the maximum number of KV pairs ($L/4$ which is the upper limit imposed by MQAR with sequence length $L$). Similarly to (Beck et al., 2024), training follows a curriculum: it begins on sequences of length 128 and every 20 epochs progresses to lengths of 256, 512, and 1024. Following prior MQAR benchmarks, the trained models consisted of two gated linear attention layers with a hidden dimension of 128 and a

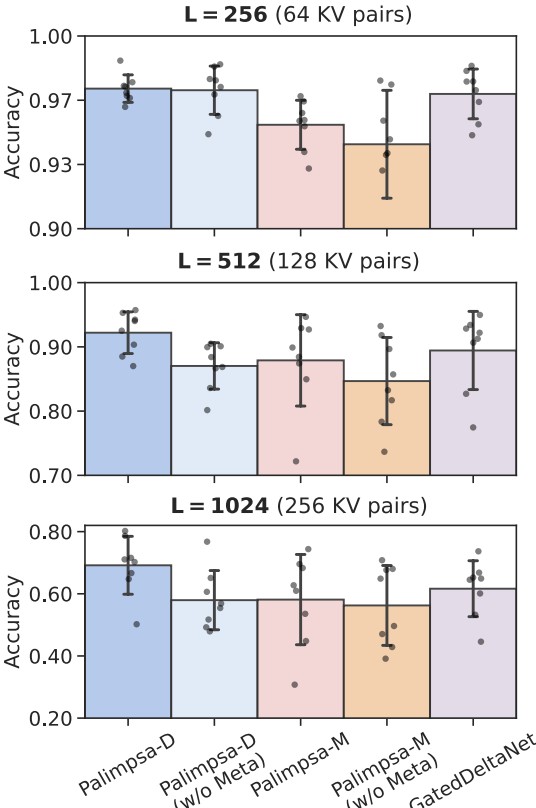

*Figure 2.* **Curriculum MQAR experiments.** Accuracy averaged over 8 seeds for the best learning rate per model. Individual run accuracies are shown as black dots; error bars represent ±1 standard deviation. Task difficulty increases with sequence length $L$. "w/o Meta" indicates that metaplasticity was disabled for those models.

value dimension expansion factor of 2. To make the task more challenging, we reduce the available state size by utilizing 8 heads. We isolate the effect of metaplasticity by evaluating Palimpsa-D and Palimpsa-M and their ablations without their metaplasticity (noting that Palimpsa-M without metaplasticity is equivalent to Mamba2, as previously explained). We include Gated Deltanet as a baseline for comparison. Results are reported over 8 runs per model using different seeds for both initialization and data generation. We explored several learning rates distributed in log space from $10^{-3}$ to $10^{-2}$, and selected the optimal learning rate for each model across seeds and sequence sizes. Results shown in Fig. 2 indicate that metaplasticity consistently improves average performance for both Palimpsa-D and Palimpsa-M. The performance gap between metaplastic and non-metaplastic variants increases with sequence length, particularly for Palimpsa-D. This suggests that increasing task complexity induces the model to leverage metaplasticity to mitigate overwriting. Overall, Palimpsa-D is the best performing model, followed by Gated Deltanet.

*Table 1.* Comparison of gated attention models from the metaplasticity perspective

| Layer | Stability Matrix | Input Gating | Forgetting | Objective resolution |
|---|---|---|---|---|
| **Longhorn** | $\mathbb{I}_{d_k}$ | $\boldsymbol{\beta}_t \in \mathbb{R}^{d_v}$ | None | Diagonal approximation |
| | *Update Rule:* $\boldsymbol{\mu}_t = (\mathbb{1}_{d_v \times d_k} - \frac{\boldsymbol{\beta}_t \otimes \boldsymbol{k}_t^2}{1 + \boldsymbol{\beta}_t \boldsymbol{k}_t^\top \boldsymbol{k}_t}) \odot \boldsymbol{\mu}_{t-1} + \frac{\boldsymbol{\beta}_t \odot \boldsymbol{v}_t}{1 + \boldsymbol{\beta}_t \boldsymbol{k}_t^\top \boldsymbol{k}_t} \otimes \boldsymbol{k}_t$ | | | |
| **Deltanet** | $\mathbb{I}_{d_k}$ | $\beta_t \in \mathbb{R}$ | None | First order approximation |
| | *Update Rule:* $\boldsymbol{\mu}_t = \boldsymbol{\mu}_{t-1}(\mathbb{I}_{d_k} - \beta_t \boldsymbol{k}_t \boldsymbol{k}_t^\top) + \beta_t \boldsymbol{v}_t \boldsymbol{k}_t^\top$ | | | |
| **Gated Deltanet** | $\mathbb{I}_{d_k}$ | $\beta_t \in \mathbb{R}$ | $\alpha_t \in \mathbb{R}$ | First order approximation |
| | *Update Rule:* $\boldsymbol{\mu}_t = \boldsymbol{\mu}_{t-1}(\alpha_t(\mathbb{I}_{d_k} - \beta_t \boldsymbol{k}_t \boldsymbol{k}_t^\top)) + \beta_t \boldsymbol{v}_t \boldsymbol{k}_t^\top$ | | | |
| **Mesa** | $\boldsymbol{\Sigma}_t^{-1}$ | $\beta_t \in \mathbb{R}$ | $\alpha_t \in \mathbb{R}$ | Conjugate grad. approx. |
| | *Update Rule:* $\boldsymbol{\Sigma}_t^{-1} = \alpha_t \boldsymbol{\Sigma}_{t-1}^{-1} + (1-\alpha_t)\boldsymbol{I}_{prior} + \beta_t \boldsymbol{k}_t \boldsymbol{k}_t^\top$ $\boldsymbol{\mu}_t = [\alpha_t \boldsymbol{\mu}_{t-1} \boldsymbol{\Sigma}_{t-1}^{-1} + \beta_t \boldsymbol{v}_t \boldsymbol{k}_t^\top]\boldsymbol{\Sigma}_t$ | | | |
| **Palimpsa** | $\text{diag}(\boldsymbol{I}_{t-1,i})$ | $\boldsymbol{\beta}_t \in \mathbb{R}^{d_v}$ | $\alpha_t \in \mathbb{R}$ | Diagonal approximation |
| | *Update Rule:* $\boldsymbol{I}_t = \alpha_t \boldsymbol{I}_{t-1} + (1-\alpha_t)\boldsymbol{I}_{prior} + \boldsymbol{\beta}_t \otimes \boldsymbol{k}_t^2,$ $\boldsymbol{\mu}_t = \alpha_t \frac{\boldsymbol{I}_{t-1}}{\boldsymbol{I}_t} \odot \boldsymbol{\mu}_{t-1} + \frac{1}{\boldsymbol{I}_t} \odot [(\boldsymbol{\beta}_t \odot \boldsymbol{v}_t) \otimes \boldsymbol{k}_t]$ | | | |
| **Palimpsa** $\left\lvert \frac{I_t - I_{prior}}{I_{prior}} \right\rvert \ll 1$ | *Update Rule:* $\boldsymbol{\mu}_t = \alpha_t \boldsymbol{\mu}_{t-1} + \frac{\boldsymbol{\beta} \odot \boldsymbol{v}_t}{I_{prior}} \otimes \boldsymbol{k}_t$ | | | ($\cong$ Mamba2, see text) |

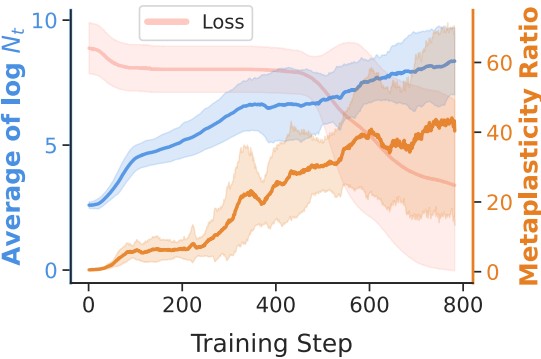

*Figure 3.* **Palimpsa's Learning Dynamics:** Memory window $N_t$ (blue), averaged over the context length, and the metaplasticity ratio (orange), defined on the final state importance as $(I_{\max} - I_{\min})/I_{\min}$, and the training loss (pink). A higher ratio indicates stronger differentiation between plastic and consolidated synapses. Shaded regions represent the standard deviation over 8 seeds.

### 3.2. Palimpsa's Learning Dynamics on MQAR

As derived in the methods section, the forgetting gate implies an effective memory window $N_t$ via the relation $\alpha_t = (1 - \frac{1}{N_t}) = e^{-Ad_t}$. In Fig. 3, we track the evolution of $N_t$ and the *metaplasticity ratio* during MQAR training, defined as $(I_{\max} - I_{\min})/I_{\min}$ (noting that $I_{min} > I_{prior} > 0$

holds). The exponential growth of $N_t$ in early epochs demonstrates that the meta-learner correctly identifies that solving MQAR requires slow forgetting (large $N_t$). When $N_t$ becomes large enough for the task, the training loss starts to decrease. This metaplasticity ratio rises to $\approx 40$, meaning highly consolidated synapses become 40-fold more stable than plastic ones. Notably, during the first 200 steps, the ratio remains low because $N_t$ is small: high forgetting prevents the accumulation of importance, keeping the model in a regime equivalent to Mamba2.

### 3.3. Language Tasks: Fineweb-Edu and Commonsense Reasoning

We pre-train Palimpsa variants, Gated Deltanets, and a Transformer on the Fineweb-edu dataset at academic scales for two different sizes, 170M and 760M parameters, and for 15B and 30B tokens, respectively. We did not include Mesanet in our benchmarks as, at the time of writing, their public code intended for GPU led to numerical instabilities and their reported results based on SlimPajama. We evaluate each pre-trained model on language modeling and Commonsense Reasoning. Following prior work (Von Oswald et al., 2023b; Liu et al., 2025; Yang et al., 2025), we test Wikitext perplexity and zero shot performance on a range of tasks such as LAMBADA (Paperno et al., 2016),

*Table 2.* Performance of Palimpsa variants and baselines on language modeling and Commonsense Reasoning tasks. The best results are in bold. The best results within a subgroup are underlined. ppl:perplexity, acc: accuracy, acc_n: normalized accuracy.

| Model | Wiki. ppl ↓ | LMB. ppl ↓ | LMB. acc ↑ | PIQA acc_n ↑ | Hella. acc_n ↑ | Wino. acc ↑ | ARC-e acc ↑ | ARC-c acc_n ↑ | SIQA acc ↑ | Avg. acc↑ |
|---|---|---|---|---|---|---|---|---|---|---|
| *170M Model Variants* | | | | | | | | | | |
| Transformer++ | 29.51 | 44.90 | **32.12** | 65.18 | 37.64 | 52.17 | 55.56 | 26.19 | 37.77 | 43.80 |
| Gated Deltanet | 29.16 | 41.87 | 30.91 | 64.91 | 38.18 | 48.86 | 56.73 | 26.37 | **39.20** | 43.59 |
| Palimpsa-D (w/o Meta) | 30.80 | 46.83 | 27.89 | 65.02 | 37.97 | 52.72 | 57.95 | 26.79 | 39.00 | 43.91 |
| Palimpsa-D (Fine-tuned 1BT) | 29.73 | 40.26 | 29.26 | 65.45 | 38.41 | 51.62 | 58.04 | 26.02 | 38.89 | 43.96 |
| Palimpsa-D (Fully Trained) | 30.81 | 42.27 | 29.46 | 65.13 | 38.02 | 50.91 | 55.85 | 25.77 | 39.00 | 43.45 |
| Palimpsa-M (w/o Meta) | 31.64 | 48.42 | 28.08 | 65.40 | 37.25 | 50.04 | 55.47 | 26.02 | 38.02 | 42.90 |
| Palimpsa-M (Fine-tuned 1BT) | 30.58 | 42.01 | 29.83 | 65.02 | 37.81 | 51.38 | 56.10 | 25.68 | 38.54 | 43.48 |
| Palimpsa-M (Fully Trained) | 31.40 | 43.02 | 30.62 | 64.91 | 38.00 | 50.28 | 56.48 | 26.71 | 38.02 | 43.57 |
| *760M Model Variants* | | | | | | | | | | |
| Transformer++ | 18.91 | 18.84 | 42.03 | 70.40 | 51.14 | 54.93 | 66.58 | 33.70 | 40.17 | 51.28 |
| Gated Deltanet | 19.06 | 17.06 | 41.74 | 71.33 | 51.74 | 55.41 | 67.30 | 32.17 | 40.84 | 51.50 |
| Palimpsa-D (w/o Meta) | 19.33 | 16.54 | 41.98 | 70.40 | 51.46 | 55.64 | 67.93 | 34.47 | 40.89 | 51.82 |
| Palimpsa-D (Fine-tuned 2BT) | 19.02 | **14.86** | 43.55 | 71.06 | 51.63 | 56.12 | 67.68 | 34.64 | 41.20 | 52.27 |
| Palimpsa-M (w/o Meta) | 19.96 | 16.34 | 42.15 | 70.73 | 50.61 | 57.06 | 67.47 | 32.76 | 39.92 | 51.53 |
| Palimpsa-M (Fine-tuned 2BT) | 19.60 | 16.35 | 42.17 | 71.06 | 50.91 | 56.99 | 67.97 | 34.39 | 41.61 | 52.16 |
| *2.7B Model Variants* | | | | | | | | | | |
| Palimpsa-M (w/o Meta) | **14.46** | **4.79** | 62.82 | 75.73 | 68.12 | 63.85 | 60.06 | 39.16 | **44.78** | 59.22 |
| Palimpsa-M (Fine-tuned 15BT) | 14.51 | **4.79** | 63.26 | 76.17 | 68.58 | 65.19 | 63.43 | 39.33 | 44.47 | **60.06** |

PIQA (Bisk et al., 2020), HellaSwag (Zellers et al., 2019), WinoGrande (Sakaguchi et al., 2021), ARC (Clark et al., 2018), and SIQA (Sap et al., 2019).

We used the Flame framework to train models from scratch (Zhang & Yang, 2025). We also fine-tuned the non-metaplastic Palimpsa variants into metaplastic ones, starting from 14BT, thereby ensuring the same token budget. At this scale, Palimpsa-D (fine-tuned) emerges as the overall winner. As a limitation, fully trained Palimpsa-D underperforms its fine-tuned counterpart. Learning metaplasticity from a random initialization is harder than starting from a checkpoint that already uses basic plasticity.

While fine-tuning benefits both architectures, the performance gain is more pronounced for Palimpsa-M. This was expected as Palimpsa-M does not have a gated MLP, and therefore incorporates more Palimpsa-M layers for the same parameter budget across all models we evaluated. Interestingly, metaplastic models demonstrate superior performance on LAMBADA perplexity. Since LAMBADA requires predicting the final word based on broad context, this improvement is consistent with the better memory retention expected from the metaplastic model.

For the 760M parameter scale, all models were trained for a total of 30B tokens. Metaplastic variants were fine-tuned for 2B tokens starting from the 28B token checkpoint of their non metaplastic counterparts to maintain training budget

parity. Since the fine-tuned model emerged as the winner for the smaller model, we concentrated our computing resources on the fine-tuning of larger metaplastic models.

The results at 760M confirm the impact of metaplastic state at medium scale: Palimpsa-D and Palimpsa-M take the top two spots. The fine-tuned metaplastic models improve upon their respective baselines by approximately 0.6 points in average accuracy, with Palimpsa-D outperforming the Gated Deltanet baseline by nearly 0.8 points (52.27 vs 51.50). Beyond average accuracy, Palimpsa expands memory capacity significantly. This is supported by the per-token NLL analysis on the 760M models (Fig. 4), where Palimpsa variants degrade more slowly than their non-metaplastic counterparts at large token positions, processing nearly $5\times$ more tokens before underperforming the Transformer.

### 3.4. Large Scale and Long Context Experiments

To evaluate Palimpsa at large scale and address long-context sequence processing, we perform a model surgery on a 2.7B parameter Mamba2 model pre-trained on 300B tokens. Because Mamba2 acts as a limiting case of Palimpsa, we can calculate the effective memory window $N_t$ for each layer. As shown in Fig. 3, the metaplasticity ratio rises with the memory window, suggesting Palimpsa should have the strongest impact on layers with the weakest forgetting. Therefore, we replace the 8 Mamba2 layers exhibiting the

*Table 3.* LongBench overall results alongside RULER Variable Tracing (VT) performance at extended context lengths for the 2.7B parameter models. Best results are in bold.

| Model | LongBench | | | | | | RULER (Variable Tracing) | | | |
|---|---|---|---|---|---|---|---|---|---|---|
| | easy | hard | short | medium | long | overall | 4k | 8k | 16k | 32k |
| Palimpsa-M (w/o Meta) | **22.4** | 17.4 | **18.9** | 19.5 | 19.4 | 19.3 | **59.6** | **49.7** | **26.0** | 14.9 |
| Palimpsa-M (Fine-tuned 15BT) | **22.4** | **18.6** | 16.1 | **23.3** | **20.4** | **20.4** | 33.5 | 24.9 | 22.5 | **24.4** |

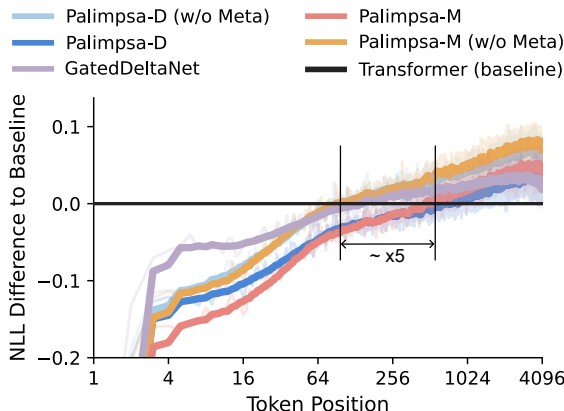

*Figure 4.* **Per-token NLL difference to a Transformer baseline on WikiText for the 760M parameter models:** Negative values indicate better performance than the Transformer at that token position. All models initially outperform the Transformer at short contexts, but degrade at longer ones. Palimpsa models (with metaplasticity) degrade more slowly, showing that metaplasticity significantly expands memory capacity.

largest $N_t$ with Palimpsa layers. We train this hybrid model using a multi-stage fine-tuning pipeline: continued pre-training for 10B tokens, context extension up to 32k tokens for 5B tokens (Gao et al., 2025; Agarwalla et al., 2024), and a final instruction fine-tuning stage (Chen et al., 2024). To guarantee a fair comparison, we concurrently applied the exact same training recipe to the Mamba2 baseline. Further details are provided in Appendix A.10.

As detailed in Table 2, this upgraded 2.7B Palimpsa-M variant outperforms the Mamba2 baseline by +0.84 points in average commonsense reasoning accuracy. This surpasses the gains observed at smaller scales (+0.58 points at 170M, and +0.63 points at 760M). For long-range dependencies (see Table 3 and Appendix A.11), Palimpsa achieves a +1.1 point overall improvement on the LongBench benchmark (Bai et al., 2025). This advantage is most pronounced in the Medium category (32k–128k context lengths), where Palimpsa dominates by nearly 4 points. Conversely, on RULER Variable Tracing (Hsieh et al., 2024) at shorter contexts (4k–16k), the metaplastic variant underperforms the Mamba2 baseline. However, Palimpsa exhibits lower degradation as sequence length increases, surpassing the baseline at 32k context lengths and confirming its superior capacity to retain information in the long-context regime.

## 4. Conclusion

We introduced Palimpsa, a Bayesian attention mechanism that reframes In-Context Learning as a continual learning problem within fixed-size memories derived from a variational free energy objective. Palimpsa implements metaplasticity: the plasticity of each of its memory states is dynamically adjusted according to a tracked importance. This approach resolves the stability-plasticity dilemma by protecting critical prior knowledge while allowing the model to forget stale information, effectively preventing both catastrophic forgetting and remembering *in-context*.

Theoretically, our framework unifies disjoint architectures, revealing that Mamba2 is a special case of Palimpsa where forgetting dominates. This insight yields a practical, scalable training pipeline: we exploit the computational speed of non-metaplastic kernels for large-scale pre-training, followed by a metaplastic fine-tuning phase, thereby enabling the use of metaplastic gated linear attention models at any scale.

Empirically, Palimpsa enables better management of state memory in constrained settings, which we show using systematic ablations of the metaplasticity. On the synthetic MQAR benchmark, Palimpsa consistently outperforms baselines, with performance gaps widening as sequence length increases. These gains translate to large-scale language modeling settings: on Commonsense Reasoning benchmarks at the 760M parameter scale, Palimpsa-D achieves an average accuracy of 52.27%, surpassing the Gated Deltanet baseline by nearly 0.8 points. Gains are particularly notable on tasks requiring broad context, such as LAMBADA, validating the model's superior capacity to retain long-range dependencies. This trend is further supported by our per-token NLL analysis (Figure 4), where Palimpsa variants degrade more slowly than their non-metaplastic counterparts as token positions grow. We demonstrate this recipe at the 2.7B parameter scale, where a surgical replacement of the 8 weakest-forgetting Mamba2 layers with Palimpsa blocks improves average commonsense reasoning accuracy by +0.84 points and overall LongBench performance by +1.1 points. This directly validates metaplasticity as a mechanism for extending the effective memory of pre-trained linear attention models without retraining from scratch.

## Impact Statement

Gated linear recurrent models offer a memory-efficient alternative to standard Transformers, making them inherently suitable for edge deployment due to their fixed-size inference state. However, this finite capacity often leads to information loss in long contexts. Palimpsa addresses this limitation by introducing Bayesian metaplasticity, which optimizes how information is learned, remembered, and forgotten within these strict memory bounds.

By enhancing the capabilities of fixed-size memory models, this work supports the deployment of performant sequence modeling on resource-constrained systems. Furthermore, Palimpsa's local update rules align well with emerging efficient hardware architectures, effectively contributing to the development of low-latency, energy-efficient intelligence at the edge.

## Code Availability

Our code is available at https://github.com/djo1996/Palimpsa.

## Acknowledgements

This work was supported by the Horizon Europe program (EIC Pathfinder METASPIN, grant number 101098651).

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

# A. Appendices

As described in the main text, Palimpsa-M builds upon Mamba2, while Palimpsa-D is based on Deltanets. These two models utilize distinct gated linear attention layers and separate backbones, allowing our metaplasticity ablation studies to evaluate two highly different points in the architectural design space.

The most striking difference is that Palimpsa-M lacks a gated MLP and relies solely on its gated linear attention layer projections for channel mixing, which is the standard configuration for Mamba2. In contrast, Palimpsa-D follows Deltanet principles by incorporating a gated MLP after the linear attention stage.

Furthermore, the models handle input gating differently: Palimpsa-M utilizes $d_t$ (a term already present in the forgetting mechanism) for input gating, whereas Palimpsa-D employs a dedicated parameter $b_t$. This allows Palimpsa-D to decorrelate forgetting and input integration. Additional distinctions include the order of the output gating and normalization at the end of the layer, and the fact that Palimpsa-M does not normalize the queries ($q_t$) and keys ($k_t$).

## A.1. Palimpsa Model Variants

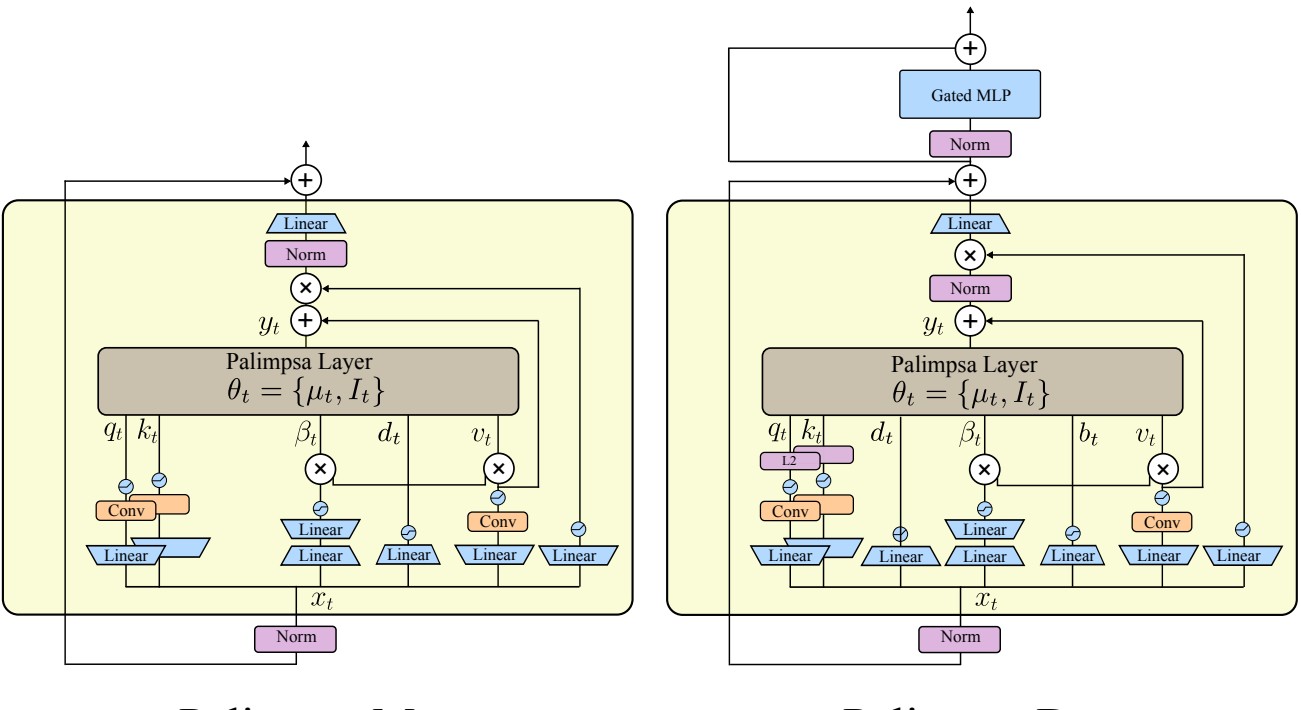

*Figure 5.* Illustration of the Palimpsa-M and Palimpsa-D architectures. While Palimpsa-M adopts the Mamba2 configuration by relying on the attention layer for channel mixing, Palimpsa-D incorporates an explicit gated MLP. Additionally, Palimpsa-D introduces a dedicated $b_t$ parameter to decorrelate input integration from the forgetting dynamics dictated by $d_t$.

## A.2. Bayesian Forgetting

Following the work of (Bonnet et al., 2025), we introduce a forgetting mechanism by considering a truncated posterior, which contains the last $N$ data points. The posterior over parameters $\boldsymbol{S}$ given data $\boldsymbol{d}_{t-N:t}$ is:

$$p(\boldsymbol{S}|\boldsymbol{d}_{t-N:t}) = \frac{p(\boldsymbol{d}_{t-N:t}|\boldsymbol{S})p(\boldsymbol{S})}{p(\boldsymbol{d}_{t-N:t})} = \frac{p(\boldsymbol{d}_t|\boldsymbol{S})p(\boldsymbol{S}|\boldsymbol{d}_{t-N:t-1})}{p(\boldsymbol{d}_t|\boldsymbol{d}_{t-N:t-1})}. \tag{6}$$

This can be expressed as a recursive update for a sliding window of size $N$. To move from the posterior over $\boldsymbol{d}_{t-N:t-1}$ to the one over $\boldsymbol{d}_{t-N+1:t}$, we incorporate the new data point $\boldsymbol{d}_t$ and remove the oldest one, $\boldsymbol{d}_{t-N}$:

$$p(\boldsymbol{S}|\boldsymbol{d}_{t-N+1:t}) \propto \underbrace{p(\boldsymbol{d}_t|\boldsymbol{S})p(\boldsymbol{S}|\boldsymbol{d}_{t-N:t-1})}_{\text{Learning}} \cdot \underbrace{\frac{1}{p(\boldsymbol{d}_{t-N}|\boldsymbol{S})}}_{\text{Forgetting}} \cdot \tag{7}$$

In principle, one may not have access at time $t$ to the likelihood of the oldest data point, $p(\boldsymbol{d}_{t-N}|\boldsymbol{S})$. As a proxy, one can use the geometric mean of the likelihoods over the previous window, $\boldsymbol{d}_{t-N:t-1}$:

$$p(\boldsymbol{d}_{t-N}, \dots, \boldsymbol{d}_{t-1}|\boldsymbol{S})^{\frac{1}{N}} \propto \left[ \frac{p(\boldsymbol{S}|\boldsymbol{d}_{t-N:t-1})}{p(\boldsymbol{S})} \right]^{\frac{1}{N}}. \tag{8}$$

This is made possible by approximating $p(\boldsymbol{S}|\boldsymbol{d}_{t-N:t-1})$, by the current variational truncated posterior $q_{\boldsymbol{\theta}_{t-1}}(\boldsymbol{S})$.

In simple terms, instead of completely discarding a single data point, this approach suggests we can discard a fraction of the joint likelihood of past data. This leads to a 'weighted' posterior where older data points have been discounted more heavily over time, keeping the total 'weight' of the posterior roughly constant (when $t \gg N$).

For Palimpsa, the forgetting needs to be input-dependent, similar to other gated recurrent models. Therefore, at each time step $t$, we discount the influence of all previous data by a factor $\frac{1}{N_t} \in [0,1]$. The resulting weighted posterior at time $t$, denoted $p_w(\boldsymbol{S}|\boldsymbol{d}_{1:t})$, is defined recursively as:

$$p_w(\boldsymbol{S}|\boldsymbol{d}_{1:t}) \propto \underbrace{p(\boldsymbol{d}_t|\boldsymbol{S})p_w(\boldsymbol{S}|\boldsymbol{d}_{1:t-1})}_{\text{Learning}} \cdot \underbrace{\left( \frac{p_w(\boldsymbol{S}|\boldsymbol{d}_{1:t-1})}{p(\boldsymbol{S})} \right)^{-\frac{1}{N_t}}}_{\text{Forgetting}}. \tag{9}$$

This is the target probability distribution that the variational distribution in Palimpsa approximates. Notably, this form of weighted posterior is the starting point for the general Bayesian framework for gated recurrent models.

### A.3. Variational Free Energy of the Weighted Posterior

We minimize the Kullback–Leibler (KL) divergence between the variational distribution $q_{\boldsymbol{\theta}}(\boldsymbol{S}_i)$ and the weighted posterior $p_w(\boldsymbol{S_i}|\boldsymbol{d}_{1:t})$ by finding the optimal parameters $\boldsymbol{\theta}_{t,i}$:

$$\boldsymbol{\theta}_{t,i} = \underset{\boldsymbol{\theta}}{\operatorname{argmin}} \, D_{\text{KL}} \left[ \, q_{\boldsymbol{\theta}}(\boldsymbol{S}_i) \, \| \, p_w(\boldsymbol{S_i}|\boldsymbol{d}_{1:t}) \right], \quad i = 1, \dots, d_v.$$

Using the definition of the KL divergence, the recursive update from the previous section (Equation 9), and assuming that the previous posterior is well-approximated by our variational distribution, $p_w(\boldsymbol{S_i}|\boldsymbol{d}_{1:t-1}) \approx q_{\boldsymbol{\theta}_{t-1,i}}(\boldsymbol{S}_i)$, we can show that this is equivalent to minimizing the variational free energy $\mathcal{F}_{t,i}$:

$$\mathcal{F}_{t,i} = D_{\text{KL}} \left[ q_{\boldsymbol{\theta}_{t,i}}(\boldsymbol{S}_i) \, \| \, q_{\boldsymbol{\theta}_{t-1,i}}(\boldsymbol{S}_i) \right] - \mathbb{E}_{q_{\boldsymbol{\theta}_{t,i}}(\boldsymbol{S}_i)} \left[ \log p(\boldsymbol{d}_t|\boldsymbol{S}_i) - \frac{1}{N_t} \log \frac{q_{\boldsymbol{\theta}_{t-1,i}}(\boldsymbol{S}_i)}{p(\boldsymbol{S}_i)} \right].$$

Now, using the standard formulas for Gaussian distributions (see A.7 for a refresher) and considering only the terms that depend on the variational mean $\boldsymbol{\mu}_i$, we obtain the free energy expression discussed in the main text:

$$\mathcal{F}_{t,i}(\boldsymbol{\mu}_i) = \underbrace{\frac{\beta_{t,i}}{2}\|\boldsymbol{\mu}_i^T \boldsymbol{k}_t - v_{t,i}\|^2}_{\text{plasticity}} + \underbrace{\frac{1}{N_t}\frac{\|\boldsymbol{\mu}_i\|^2}{2\sigma_{prior}^2}}_{\text{forgetting}} + \underbrace{\frac{1}{2}\left(1 - \frac{1}{N_t}\right)(\boldsymbol{\mu}_{t-1,i} - \boldsymbol{\mu}_i)^T \boldsymbol{\Sigma}_{t-1,i}^{-1}(\boldsymbol{\mu}_{t-1,i} - \boldsymbol{\mu}_i)}_{\text{stability}}.$$

For Palimpsa we are also interested in the terms that depend on $\boldsymbol{\Sigma}_{t,i}$. First, let's derive the gradient with respect to all variational parameters.

**Calculating the Gradient of $\mathcal{F}_{t,i}$:** For readability, we first define the cost term $\mathcal{C}_{t,i}$ as the expected negative log-likelihood:

$$\mathcal{C}_{t,i} := -\mathbb{E}_{q_{\boldsymbol{\theta}_i}(\boldsymbol{S}_i)} \left[ \log p(v_{t,i}|\boldsymbol{k}_t, \beta_{t,i}, \boldsymbol{S}_i) \right], \tag{10}$$

To compute the partial derivatives of $\mathcal{C}_{t,i}$, we use the reparameterization trick: $\boldsymbol{S}_i = \boldsymbol{\mu}_i + \boldsymbol{A}_i\boldsymbol{\epsilon}_i$, and $\boldsymbol{A}_i\boldsymbol{A}_i^\top = \boldsymbol{\Sigma}_{t,i}$, where

$\epsilon_i \sim \mathcal{N}(0, \mathbb{I})$. The cost function can now be written as follows:

$$\mathcal{C}_{t,i} = \mathbb{E}_{\epsilon_i} \left[ \mathcal{L}_i(\boldsymbol{S}_i) \right] \tag{11}$$

$$\frac{\partial \mathcal{C}_i}{\partial \boldsymbol{\mu}_i} = \mathbb{E}_{\epsilon_i} \left[ \frac{\partial \mathcal{L}_i(\boldsymbol{S}_i)}{\partial \boldsymbol{S}_i} \right] \qquad \frac{\partial \mathcal{C}_i}{\partial \boldsymbol{A}_i} = \mathbb{E}_{\epsilon_i} \left[ \frac{\partial \mathcal{L}_i(\boldsymbol{S}_i)}{\partial \boldsymbol{S}_i} \epsilon_i^\top \right]. \tag{12}$$

Generally, these gradients would be estimated using Monte Carlo sampling. However, in Palimpsa, ICL amounts to a linear regression problem so the gradients can be calculated analytically. Substituting the derivative of the linear regression log-likelihood gives:

$$\frac{\partial \mathcal{C}_i}{\partial \boldsymbol{\mu}_i} = \mathbb{E}_{\epsilon_i} \left[ \beta_{t,i} \left( (\boldsymbol{\mu}_i + \boldsymbol{A}_i \epsilon_i)^\top \boldsymbol{k}_t - v_{t,i} \right) \boldsymbol{k}_t \right] \qquad \frac{\partial \mathcal{C}_i}{\partial \boldsymbol{A}_i} = \mathbb{E}_{\epsilon_i} \left[ \beta_{t,i} \left( (\boldsymbol{\mu}_i + \boldsymbol{A}_i \epsilon_i)^\top \boldsymbol{k}_t - v_{t,i} \right) \boldsymbol{k_t} \epsilon_i^\top \right]. \tag{13}$$

Knowing that $\mathbb{E}_{\epsilon_i}[\epsilon_i] = \vec{0}$ and $\mathbb{E}_{\epsilon_i}[\epsilon_i \epsilon_i^\top] = \mathbb{I}$ (the identity matrix), we can resolve the expectations to get the final analytical gradients of the cost term:

$$\frac{\partial \mathcal{C}_i}{\partial \boldsymbol{\mu}_i} = \beta_{t,i} \left( \boldsymbol{k}_t \boldsymbol{k}_t^\top \boldsymbol{\mu}_i - v_{t,i} \boldsymbol{k}_t \right) \qquad \frac{\partial \mathcal{C}_i}{\partial \boldsymbol{A}_i} = \beta_{t,i} \boldsymbol{k}_t \boldsymbol{k}_t^\top \boldsymbol{A}_i. \tag{14}$$

Finally, we add the gradients from the other terms in the free energy objective (the KL-divergence and forgetting terms) to obtain the full gradients of $\mathcal{F}_{t,i}$:

$$\frac{\partial \mathcal{F}_{t,i}}{\partial \boldsymbol{\mu}_i} = \left( 1 - \frac{1}{N_t} \right) \boldsymbol{\Sigma}_{t-1,i}^{-1} (\boldsymbol{\mu}_i - \boldsymbol{\mu}_{t-1,i}) + \frac{1}{N_t} \boldsymbol{\mu}_i + \beta_{t,i} \left( \boldsymbol{k}_t \boldsymbol{k}_t^\top \boldsymbol{\mu}_i - v_{t,i} \boldsymbol{k}_t \right), \tag{15}$$

$$\frac{\partial \mathcal{F}_{t,i}}{\partial \boldsymbol{A}_i} = \left[ \left( 1 - \frac{1}{N_t} \right) \boldsymbol{\Sigma}_{t-1,i}^{-1} + \frac{1}{N_t} \boldsymbol{I}_{prior} \right] \boldsymbol{A}_i - (\boldsymbol{A}_i^{-1})^\top + \beta_{t,i} \boldsymbol{k}_t \boldsymbol{k}_t^\top \boldsymbol{A}_i. \tag{16}$$

**Closed-Form Solution of $\mathcal{F}_{t,i}$:** To find the optimal variational parameters, we set the gradients of the free energy $\mathcal{F}_{t,i}$ to zero and solve. First, to find the optimal covariance matrix $\boldsymbol{\Sigma}_{t,i}$, we solve for $\boldsymbol{A}_{t,i}$ in the equation:

$$\frac{\partial \mathcal{F}_{t,i}}{\partial \boldsymbol{A}_{t,i}} = \boldsymbol{0}. \tag{17}$$

Setting the gradient expression from the previous section to zero gives:

$$\left[ \left( 1 - \frac{1}{N_t} \right) \boldsymbol{\Sigma}_{t-1,i}^{-1} + \frac{1}{N_t} \boldsymbol{I}_{prior} + \beta_{t,i} \boldsymbol{k}_t \boldsymbol{k}_t^\top \right] \boldsymbol{A}_{t,i} - (\boldsymbol{A}_{t,i}^{-1})^\top = \boldsymbol{0}. \tag{18}$$

To solve for the full covariance matrix $\boldsymbol{\Sigma}_{t,i} = \boldsymbol{A}_{t,i} \boldsymbol{A}_{t,i}^\top$, we can multiply on the right by $\boldsymbol{A}_{t,i}^\top$:

$$\left[ \left( 1 - \frac{1}{N_t} \right) \boldsymbol{\Sigma}_{t-1,i}^{-1} + \frac{1}{N_t} \boldsymbol{I}_{prior} + \beta_{t,i} \boldsymbol{k}_t \boldsymbol{k}_t^\top \right] \boldsymbol{\Sigma}_{t,i} - \boldsymbol{I} = \boldsymbol{0}. \tag{19}$$

Rearranging the terms, we find that the new precision matrix (the inverse covariance) is a sum of the discounted old precision, the prior precision, and the new data term:

$$\boldsymbol{\Sigma}_{t,i}^{-1} = \left( 1 - \frac{1}{N_t} \right) \boldsymbol{\Sigma}_{t-1,i}^{-1} + \frac{1}{N_t} \boldsymbol{I}_{prior} + \beta_{t,i} \boldsymbol{k}_t \boldsymbol{k}_t^\top. \tag{20}$$

The closed-form update for the covariance is the inverse of this expression:

$$\boxed{ \boldsymbol{\Sigma}_{t,i} = \left[ \left( 1 - \frac{1}{N_t} \right) \boldsymbol{\Sigma}_{t-1,i}^{-1} + \frac{1}{N_t} \boldsymbol{I}_{prior} + \beta_{t,i} \boldsymbol{k}_t \boldsymbol{k}_t^\top \right]^{-1} } \tag{21}$$

As discussed in the main text, the major challenge in this analytical approach is to compute the $d_k \times d_k$ matrix inverse at each time step. In the case without forgetting ($N \to \infty$), the prior term vanishes, and the precision matrix update is a pure

rank-1 addition: $\boldsymbol{\Sigma}_{t,i}^{-1} = \boldsymbol{\Sigma}_{t-1,i}^{-1} + \beta_{t,i}\boldsymbol{k}_t\boldsymbol{k}_t^\top$. In this regime, the covariance matrix can be recursively computed without direct matrix inversion using the standard Sherman-Morrison formula:

$$\boldsymbol{\Sigma}_{t,i} = \boldsymbol{\Sigma}_{t-1,i} - \frac{\beta_{t,i}}{1 + \beta_{t,i}\boldsymbol{k}_t^\top\boldsymbol{\Sigma}_{t-1,i}\boldsymbol{k}_t}\boldsymbol{\Sigma}_{t-1,i}\boldsymbol{k}_t\boldsymbol{k}_t^\top\boldsymbol{\Sigma}_{t-1,i}. \tag{22}$$

If we reintroduce forgetting ($N_t < \infty$) the precision update consists of rank-$d_k$ update, $\boldsymbol{I}_{prior}$ being typically the identity matrix here. The full covariance matrix cannot, therefore, be computed exactly using Sherman-Morrison formula.

### A.4. Derivation of Mesanet

In the case without forgetting ($N \to \infty$), (Von Oswald et al., 2023b) use the Sherman-Morrison formula to compute it recursively, but this scaled poorly. Another approach is to use a parallel conjugate gradient method to solve the associated linear systems, as done in Mesanet (Von Oswald et al., 2025). To connect this with the notation in Mesanet (Von Oswald et al., 2025), our precision matrix $\boldsymbol{\Sigma}_t^{-1}$ corresponds to their $\boldsymbol{H}_t + \boldsymbol{\Lambda}$, and our precision-weighted mean $\boldsymbol{\mu}_t\boldsymbol{\Sigma}_t^{-1}$ corresponds to their $\boldsymbol{G}_t$. In their work, the precision matrix is identical for all rows $i$ (i.e., $\boldsymbol{\Sigma}_{t,i}^{-1} = \boldsymbol{\Sigma}_t^{-1}$) because their $\beta_t$ is a scalar.

Similarly, we find the optimal mean $\boldsymbol{\mu}_{t,i}$ by setting its corresponding gradient to zero:

$$\frac{\partial \mathcal{F}_{t,i}}{\partial \boldsymbol{\mu}_{t,i}} = \vec{0}. \tag{23}$$

$$\left(1 - \frac{1}{N_t}\right)\boldsymbol{\Sigma}_{t-1}^{-1}(\boldsymbol{\mu}_{t,i} - \boldsymbol{\mu}_{t-1,i}) + \frac{1}{N_t}\boldsymbol{I}_{prior}\boldsymbol{\mu}_{t,i} + \beta_t\left(\boldsymbol{k}_t\boldsymbol{k}_t^\top\boldsymbol{\mu}_{t,i} - v_{t,i}\boldsymbol{k}_t\right) = \vec{0}. \tag{24}$$

Grouping the terms with $\boldsymbol{\mu}_{t,i}$ yields:

$$\left[\left(1 - \frac{1}{N_t}\right)\boldsymbol{\Sigma}_{t-1}^{-1} + \frac{1}{N_t}\boldsymbol{I}_{prior} + \beta_t\boldsymbol{k}_t\boldsymbol{k}_t^\top\right]\boldsymbol{\mu}_{t,i} = \left(1 - \frac{1}{N_t}\right)\boldsymbol{\Sigma}_{t-1}^{-1}\boldsymbol{\mu}_{t-1,i} + \beta_t v_{t,i}\boldsymbol{k}_t. \tag{25}$$

Recognizing the term in brackets as the new precision $\boldsymbol{\Sigma}_t^{-1}$, we arrive at the solution for the row-mean:

$$\boldsymbol{\mu}_{t,i} = \boldsymbol{\Sigma}_t\left[\left(1 - \frac{1}{N_t}\right)\boldsymbol{\Sigma}_{t-1}^{-1}\boldsymbol{\mu}_{t-1,i} + \beta_t v_{t,i}\boldsymbol{k}_t\right] \tag{26}$$

By defining the forgetting factor $\alpha_t = \left(1 - \frac{1}{N_t}\right)$ and stacking the transposed row vectors $\boldsymbol{\mu}_{t,i}^\top$ into the full state matrix $\boldsymbol{\mu}_t \in \mathbb{R}^{d_v \times d_k}$, we can write the final update in matrix form. Since the precision and covariance matrices are symmetric ($\boldsymbol{\Sigma}^\top = \boldsymbol{\Sigma}$), transposing the equation shifts these matrices to the right, ensuring the dimensions properly align:

$$\boxed{\boldsymbol{\mu}_t = \left[\alpha_t\boldsymbol{\mu}_{t-1}\boldsymbol{\Sigma}_{t-1}^{-1} + \beta_t\boldsymbol{v}_t\boldsymbol{k}_t^\top\right]\boldsymbol{\Sigma}_t} \tag{27}$$

### A.5. Derivation of Palimpsa

In Palimpsa, we aim to solve the update equations with a vector input gating $\boldsymbol{\beta}_t \in \mathbb{R}^{d_v}$, where each row of states has its own rate. This change prevents a simple closed-form solution for the matrix inverse. To make the problem tractable and computationally efficient, we introduce a diagonal approximation for the precision matrix. By assuming the precision matrices are diagonal, $\boldsymbol{\Sigma}_{t,i}^{-1} = \mathrm{diag}(\boldsymbol{I}_{t,i})$, we can apply this approximation to the closed-form solutions from the previous section. This yields element-wise update rules for the diagonal precision vector $\boldsymbol{I}_{t,i}$ and the mean vector $\boldsymbol{\mu}_{t,i}$ for each row $i$:

$$\boldsymbol{I}_{t,i} = \left(1 - \frac{1}{N_t}\right)\boldsymbol{I}_{t-1,i} + \frac{1}{N_t}\boldsymbol{I}_{prior} + \beta_{t,i}\boldsymbol{k}_t^2 \tag{28}$$

$$\boldsymbol{\mu}_{t,i} = \left(1 - \frac{1}{N_t}\right)\frac{\boldsymbol{I}_{t-1,i}}{\boldsymbol{I}_{t,i}} \odot \boldsymbol{\mu}_{t-1,i} + \frac{1}{\boldsymbol{I}_{t,i}} \odot (\beta_{t,i}v_{t,i}\boldsymbol{k}_t) \tag{29}$$

where $\boldsymbol{k}_t^2$ denotes the element-wise square of $\boldsymbol{k}_t$, and all divisions are element-wise. Note that this is a stronger approximation than a standard mean-field (diagonal covariance) assumption because we derived the fully-coupled solution first and only then discarded the off-diagonal terms. This approach is similar to the diagonal approximation used in Longhorn (Liu et al., 2025) and is equivalent to treating each parameter (or "synapse") as an independent Gaussian distribution.

From here, by defining the forgetting factor $\alpha_t = (1 - \frac{1}{N_t})$ and stacking the row vectors into matrices, we can write the final updates for Palimpsa in matrix form:

$$\boxed{\boldsymbol{I}_t = \alpha_t \boldsymbol{I}_{t-1} + (1 - \alpha_t)\boldsymbol{I}_{prior} + \boldsymbol{\beta}_t \otimes \boldsymbol{k}_t^2} \tag{30}$$

$$\boxed{\boldsymbol{\mu}_t = \alpha_t \frac{\boldsymbol{I}_{t-1}}{\boldsymbol{I}_t} \odot \boldsymbol{\mu}_{t-1} + \frac{1}{\boldsymbol{I}_t} \odot \left[(\boldsymbol{\beta}_t \odot \boldsymbol{v}_t) \otimes \boldsymbol{k}_t\right]} \tag{31}$$

## A.6. Derivation of Deltanet and Gated Deltanet

Gated Deltanet can be derived similarly by suppressing the forgetting for the standard Deltanet. Starting from the free energy equation and taking $\beta_t \in \mathbb{R}$:

$$\mathcal{F}_{t,i}(\boldsymbol{\mu}_i) = \underbrace{\frac{\beta_t}{2}\|\boldsymbol{\mu}_i^T \boldsymbol{k}_t - v_{t,i}\|^2}_{\text{plasticity}} + \underbrace{\frac{1}{N_t}\frac{\|\boldsymbol{\mu}_i\|^2}{2\sigma_{prior}^2}}_{\text{forgetting}} + \underbrace{\frac{1}{2}\left(1 - \frac{1}{N_t}\right)(\boldsymbol{\mu}_{t-1,i} - \boldsymbol{\mu}_i)^T \boldsymbol{\Sigma}_{t-1,i}^{-1}(\boldsymbol{\mu}_{t-1,i} - \boldsymbol{\mu}_i)}_{\text{stability}}.$$

with the simplifications $\boldsymbol{\Sigma}_{t-1,i}^{-1} = \mathbb{I}_{d_k}$ and $I_{prior} = 1$. If we consider the plasticity term of the loss to be linear around $\boldsymbol{\mu}_{t-1,i}$ (a first-order approximation), we assume that:

$$\beta_t \left(\boldsymbol{k}_t \boldsymbol{k}_t^\top \boldsymbol{\mu}_i - v_{t,i}\boldsymbol{k}_t\right) \approx \beta_t \left(\boldsymbol{k}_t \boldsymbol{k}_t^\top \boldsymbol{\mu}_{t-1,i} - v_{t,i}\boldsymbol{k}_t\right).$$

Then, setting the full gradient to zero, $\frac{\partial \mathcal{F}_{t,i}}{\partial \boldsymbol{\mu}_i} = \vec{0}$, is given by:

$$\left(1 - \frac{1}{N_t}\right)\mathbb{I}_{d_k}(\boldsymbol{\mu}_i - \boldsymbol{\mu}_{t-1,i}) + \frac{1}{N_t}\boldsymbol{\mu}_i + \beta_t \left(\boldsymbol{k}_t \boldsymbol{k}_t^\top \boldsymbol{\mu}_{t-1,i} - v_{t,i}\boldsymbol{k}_t\right) = \vec{0}. \tag{32}$$

Solving for $\boldsymbol{\mu}_i$ yields:

$$\boldsymbol{\mu}_i = \left(1 - \frac{1}{N_t}\right)\boldsymbol{\mu}_{t-1,i} - \beta_t \left(\boldsymbol{k}_t \boldsymbol{k}_t^\top \boldsymbol{\mu}_{t-1,i} - v_{t,i}\boldsymbol{k}_t\right). \tag{33}$$

From there, by defining $\alpha_t = (1 - \frac{1}{N_t})$, we can write the final solution in matrix form as:

$$\boldsymbol{\mu}_t = \boldsymbol{\mu}_{t-1}\left(\alpha_t \mathbb{I}_{d_k} - \beta_t \boldsymbol{k}_t \boldsymbol{k}_t^\top\right) + \beta_t \boldsymbol{v}_t \boldsymbol{k}_t^\top. \tag{34}$$

This is equivalent to Gated Deltanet with the right reparametrization, and the same as the standard Deltanet when $N \to \infty$, i.e., $\alpha_t \to 1$.

## A.7. Gaussian Cheat Sheet

**Entropy**   The entropy of a multivariate Gaussian distribution $q_{\boldsymbol{\theta}_1}(\boldsymbol{S}_i) = \mathcal{N}(\boldsymbol{\mu}_1, \boldsymbol{\Sigma}_1)$ of dimension $d_k$ is given by:

$$H(q_{\boldsymbol{\theta}_1}) = -\mathbb{E}_{q_{\boldsymbol{\theta}_1}(\boldsymbol{S}_i)}\left[\log q_{\boldsymbol{\theta}_1}(\boldsymbol{S}_i)\right]$$

$$H(q_{\boldsymbol{\theta}_1}) = \frac{d_k}{2}\log(2\pi e) + \frac{1}{2}\log\det(\boldsymbol{\Sigma}_1)$$

**KL Divergence**   The KL divergence between two multivariate Gaussian distributions $q_{\boldsymbol{\theta}_1}(\boldsymbol{S}_i) = \mathcal{N}(\boldsymbol{\mu}_1, \boldsymbol{\Sigma}_1)$ and $q_{\boldsymbol{\theta}_2}(\boldsymbol{S}_i) = \mathcal{N}(\boldsymbol{\mu}_2, \boldsymbol{\Sigma}_2)$ is given by:

$$D_{\text{KL}}\left[q_{\boldsymbol{\theta}_1}(\boldsymbol{S}_i) \,\|\, q_{\boldsymbol{\theta}_2}(\boldsymbol{S}_i)\right] = \mathbb{E}_{q_{\boldsymbol{\theta}_1}(\boldsymbol{S}_i)}\left[\log \frac{q_{\boldsymbol{\theta}_1}(\boldsymbol{S}_i)}{q_{\boldsymbol{\theta}_2}(\boldsymbol{S}_i)}\right]$$

$$D_{\text{KL}}\left[q_{\boldsymbol{\theta}_1}(\boldsymbol{S}_i) \,\|\, q_{\boldsymbol{\theta}_2}(\boldsymbol{S}_i)\right] = \frac{1}{2}\left[\text{tr}(\boldsymbol{\Sigma}_2^{-1}\boldsymbol{\Sigma}_1) + (\boldsymbol{\mu}_2 - \boldsymbol{\mu}_1)^T\boldsymbol{\Sigma}_2^{-1}(\boldsymbol{\mu}_2 - \boldsymbol{\mu}_1) - d_k + \log\frac{\det\boldsymbol{\Sigma}_2}{\det\boldsymbol{\Sigma}_1}\right]$$

**Cross-Entropy**   The cross-entropy between two multivariate Gaussian distributions $q_{\boldsymbol{\theta}_1}(\boldsymbol{S}_i) = \mathcal{N}(\boldsymbol{\mu}_1, \boldsymbol{\Sigma}_1)$ and $q_{\boldsymbol{\theta}_2}(\boldsymbol{S}_i) = \mathcal{N}(\boldsymbol{\mu}_2, \boldsymbol{\Sigma}_2)$ is given by:

$$H(q_{\boldsymbol{\theta}_1}, q_{\boldsymbol{\theta}_2}) = -\mathbb{E}_{q_{\boldsymbol{\theta}_1}(\boldsymbol{S}_i)}\left[\log q_{\boldsymbol{\theta}_2}(\boldsymbol{S}_i)\right]$$

It can be found using the relation:

$$H(q_{\boldsymbol{\theta}_1}, q_{\boldsymbol{\theta}_2}) = H(q_{\boldsymbol{\theta}_1}) + D_{\mathrm{KL}} \left[ q_{\boldsymbol{\theta}_1}(\boldsymbol{S}_i) \,\|\, q_{\boldsymbol{\theta}_2}(\boldsymbol{S}_i) \right]$$

The final expression is:

$$H(q_{\boldsymbol{\theta}_1}, q_{\boldsymbol{\theta}_2}) = \frac{1}{2} \left[ \mathrm{tr}(\boldsymbol{\Sigma}_2^{-1} \boldsymbol{\Sigma}_1) + (\boldsymbol{\mu}_2 - \boldsymbol{\mu}_1)^T \boldsymbol{\Sigma}_2^{-1} (\boldsymbol{\mu}_2 - \boldsymbol{\mu}_1) + d_k \log(2\pi) + \log \det \boldsymbol{\Sigma}_2 \right]$$

### A.8. MQAR Experiments

For the Multi-Query Associative Recall (MQAR) task, we employ a progressive curriculum learning strategy across four stages of increasing complexity. Models are trained sequentially on sequence lengths $L \in \{128, 256, 512, 1024\}$ with a corresponding number of key-value pairs $N_{kv} \in \{32, 64, 128, 256\}$. We perform a grid search over learning rates and execute each configuration across 8 independent random seeds to ensure statistical robustness.

The recurrence dynamics are initialized to favor long-term memory retention. Specifically, the decay parameters $\mathbf{A}$ are initialized using a linear ramp $\mathcal{U}(0.01, 0.16)$ across heads. The $d_t$ bias is initialized via a linear ramp in log-space between $0.001$ and $0.1$. Full hyperparameters are detailed in Table 4.

*Table 4.* Hyperparameters for the MQAR curriculum experiments. Initializations for $\mathbf{A}$ and $d_t$ are configured to favor long-term memory retention.

| Hyperparameter | Value / Search Space |
| --- | --- |
| Vocabulary size | 8,192 |
| Embedding dimension ($d_{model}$) | 128 |
| Number of layers | 2 |
| Number of heads ($n_{heads}$) | 8 |
| State dimension ($d_{state}$) | 16 |
| Value expansion ($E_v$) | 2 |
| Key expansion ($E_k$) | 1 |
| $\beta$ step rank | $d_{model}/16 = 8$ |
| $\mathbf{A}$ initialization | linspace$(0.01, 0.16)$ |
| $\Delta t$ initialization | logspace$(10^{-3}, 10^{-1})$ |
| Curriculum stages ($L, N_{kv}$) | (128, 32), (256, 64), (512, 128), (1024, 256) |
| Epochs per stage | 20 |
| Batch size | 128 (for $L \leq 512$), 64 (for $L = 1024$) |
| Optimizer | AdamW |
| Learning rate | [1e-3, 2.15e-3, 4.64e-3, 1e-2] |
| Random seeds | $\{1, 2, 3, 4, 5, 6, 7, 8\}$ |

### A.9. Language Modelling Experiments

We evaluate Palimpsa across two primary model scales: 170M and 760M parameters. For all models, word embeddings are tied with the language modeling head. The decay parameters $\mathbf{A}$ are initialized by sampling from a uniform distribution $\mathcal{U}(0, 16)$. For discretization, the $\Delta$ bias is initialized via a log-space ramp between $0.001$ and $0.1$.

Convolutional and linear layers are initialized with a Gaussian distribution ($\sigma = 0.02$). We utilize a cosine learning rate scheduler with a 2,000-step warmup for the initial training phase. The peak learning rates are set to $3 \times 10^{-3}$ for the 170M models and $1.25 \times 10^{-3}$ for the 760M models. Training is performed on the FineWeb-Edu dataset using the Flame framework (Zhang & Yang, 2025).

We distinguish the initialisation between standard training ($b_{scale} = 1$) and a fine-tuning regime ($b_{scale} \in [0.1, 1.0]$). In the fine-tuning stage, models resume from intermediate checkpoints with a shortened 200-step warmup. To ensure stability during this phase, the peak learning rate is reduced to $1/10$ of the original value for both model scales. Models are trained for the final 1B (15B total budget) or 2B (30B total budget) tokens, maintaining a fixed computational budget across all experiments.

*Table 5.* Architectural and training details for language modeling experiments. Palimpsa-D and Palimpsa-M utilize consistent initialization schemes. Fine-tuning runs start from the corresponding pre-trained checkpoint.

| Size | Model | Layers | Dim | Heads | $E_k$ | $E_v$ | Peak LR |
|------|-------|--------|-----|-------|-------|-------|---------|
| 170M | Transformer++ | 20 | 768 | 16 | – | – | |
| | Gated Deltanet | 19 | 768 | 16 | 1.0 | 2.0 | 3.0e-3 |
| | Palimpsa-D | 19 | 768 | 16 | 1.0 | 2.0 | |
| | Palimpsa-M | 30 | 768 | 16 | 1.0 | 2.0 | |
| 760M | Transformer++ | 25 | 1536 | 16 | – | – | |
| | Gated Deltanet | 19 | 1536 | 16 | 1.0 | 1.0 | 1.25e-3 |
| | Palimpsa-D | 19 | 1536 | 16 | 1.0 | 2.0 | |
| | Palimpsa-M | 38 | 1536 | 16 | 1.0 | 2.0 | |

### A.10. Fine-Tuning Procedure for the 2.7B Model

For our large-scale experiments, we utilize the official `state-spaces/mamba2-2.7b` checkpoint as our base model. This architecture consists of 64 layers with a hidden dimension ($d_{model}$) of 2560 and a vocabulary size of 50,277. All undefined hyperparameters fall back to the default Mamba2 configuration.

As introduced in the main text, we analyze the layer-wise effective memory window $N_t$ of the pre-trained Mamba2 model to determine the optimal placement for metaplasticity (Figure 6). We surgically replace the top 8 layers exhibiting the largest median $N_t$ across the sequence dimension (i.e., the weakest forgetting) with trainable Palimpsa blocks, preserving the Mamba2 architecture for the remaining 56 layers.

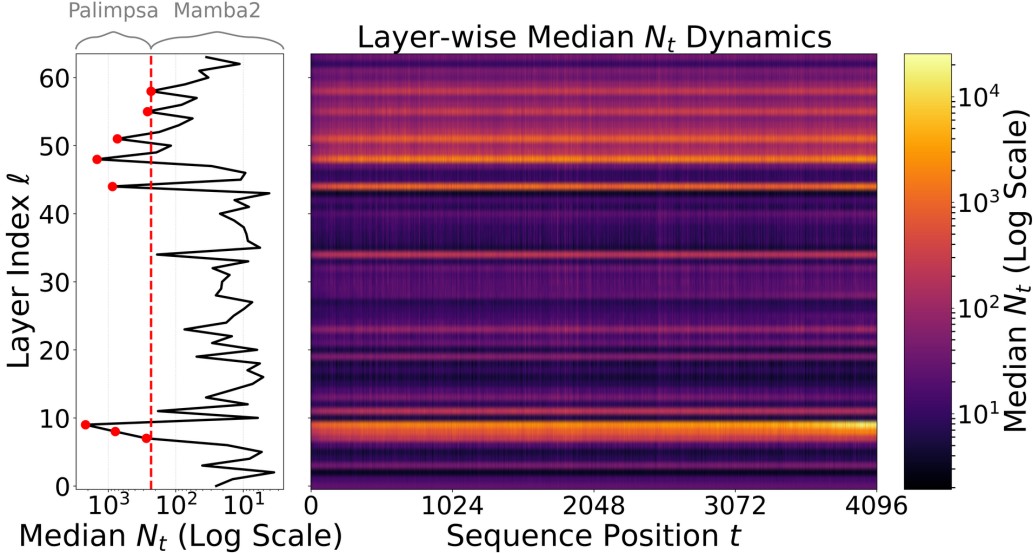

*Figure 6.* **Layer-wise Memory Window** ($N_t$): Median $N_t$ computed across heads for each layer as a function of token position. The 8 layers exhibiting the highest sustained $N_t$ (weakest forgetting, marked in red on the left panel) were selected for surgical replacement with Palimpsa layers.

After this substitution, we employ a three-stage training pipeline to adapt the two models for extended context lengths:

- **Stage 1: Continued Pre-training.** We trained the hybrid model for an additional 10B tokens on a mixture of FineWebEdu and StackV2.

- **Stage 2: Context Extension.** We extended the context window up to 32k tokens by training for 5B tokens on long-form text (Gao et al., 2025; Agarwalla et al., 2024). This dataset comprised the "arxiv" and "book" subsets of SlimPajamas, repository-level concatenations of StackV2 (C++ and Python code), and synthetic Needle-in-a-Haystack data to adapt the state to the extended time horizon.

- **Stage 3: Instruction Fine-Tuning.** Finally, the model was instruction fine-tuned on the LongAlpaca dataset (Chen et al., 2024) to align its outputs for downstream benchmark evaluations.

### A.11. Ruler Benchmark Experiments

We present the full RULER (Hsieh et al., 2024) evaluation for our 2.7B parameter models in Table 6. RULER assesses distinct aspects of long-context performances.

In the Retrieval category, a clear divergence emerges. The non-metaplastic baseline dominates single-needle retrieval (N-1, N-2, N-3), perhaps because input selectivity can effectively isolate and protects a single vital piece of information. However, the fine-tuned Palimpsa variant consistently wins when retrieving multiple distinct keys (N-MQ), multiple values sharing a key (N-MV), or ignoring hard distractors (N-MK). This suggests metaplasticity provides a superior compromise between learning and remembering when multiple contextual pieces are important.

Crucially, the Variable Tracing (VT) results perfectly illustrate our core hypothesis. While the metaplastic 2.7B variant underperforms the baseline at shorter contexts (4k–16k), it exhibits a significantly lower degradation rate as sequence length increases. By the 32k context length, Palimpsa decisively surpasses the baseline (24.4 vs. 14.9), confirming its superior capacity to retain sequential dependencies in the extended regime.

For Aggregation tasks (CWE, FWE), the baseline retains an advantage. Metaplasticity appears to negatively impact the extraction of frequent words, though the exact mechanistic reason remains an open question. Finally, results on Question Answering (HotPot, SQA) remain mixed depending on the specific context length.

*Table 6.* Comprehensive RULER Benchmark evaluation of the surgically upgraded 2.7B parameter Palimpsa-M models across context lengths from 4k to 32k. Tasks are grouped by category to isolate specific memory behaviors. Best results are in bold.

| Model | N-1 | N-2 | N-3 | Retrieval N-MQ | N-MV | N-MK | Tracing VT | Aggregation CWE | FWE | QA HotPot | SQA |
|---|---|---|---|---|---|---|---|---|---|---|---|
| | | | | 4k Context | | | | | | | |
| Palimpsa-M (w/o Meta) | **100.0** | **88.0** | **83.6** | 53.1 | 51.5 | 25.0 | **59.6** | **30.5** | **64.0** | 25.8 | **28.8** |
| Palimpsa-M (Fine-tuned 15BT) | **100.0** | 84.4 | 77.2 | **57.5** | **59.9** | **28.4** | 33.5 | 29.6 | 58.4 | **29.8** | 28.4 |
| | | | | 8k Context | | | | | | | |
| Palimpsa-M (w/o Meta) | **100.0** | **46.2** | **42.4** | 38.8 | 34.9 | **24.6** | **49.7** | **20.5** | **64.0** | **24.4** | 19.0 |
| Palimpsa-M (Fine-tuned 15BT) | **100.0** | 44.8 | **42.4** | **39.4** | **37.2** | 21.8 | 24.9 | 13.0 | 56.7 | 23.8 | **19.2** |
| | | | | 16k Context | | | | | | | |
| Palimpsa-M (w/o Meta) | **100.0** | 9.4 | **13.6** | 7.0 | 6.0 | **8.0** | **26.0** | **2.4** | **59.6** | **18.4** | 15.6 |
| Palimpsa-M (Fine-tuned 15BT) | 99.4 | **11.2** | 12.6 | **8.2** | **9.7** | **8.0** | 22.5 | 0.6 | 43.8 | 17.0 | **17.1** |
| | | | | 32k Context | | | | | | | |
| Palimpsa-M (w/o Meta) | **94.4** | 7.0 | **9.4** | 2.3 | 1.1 | 7.2 | 14.9 | **0.4** | **31.9** | 20.2 | 14.7 |
| Palimpsa-M (Fine-tuned 15BT) | 93.0 | **7.6** | 8.2 | **2.8** | **1.4** | **7.6** | **24.4** | 0.2 | 24.7 | **20.6** | **15.1** |

### A.12. Palimpsa Implementation and Parallelization

Our parallel training algorithm is implemented as a fused Triton kernel utilizing a chunk-wise associative scan. The core logic is structured as follows:

- **Chunk-wise Processing:** The input sequence is partitioned into chunks of size $B_T$.

- **Intra-Chunk Parallel Scan:** Within each chunk, the recursive state updates for the first moment $M_t$ and importance $I_t$ are computed in parallel using an associative scan.

- **Inter-Chunk Sequential Update:** The final state of one chunk is carried over as the initial state for the subsequent chunk to ensure global temporal consistency.

This approach maximizes GPU utilization by transforming the sequential recurrence into a parallelizable prefix-sum operation. The kernel also performs state checkpointing to maintain numerical stability during the backward pass.

To implement the parallel scan, we define the state as a tuple $(M_t, \bar{I}_t, A_t)$, where $\bar{I}_t = I_t - I_{\text{prior}}$ represents the centered importance. Let $t \in [1, B_T]$ be the index within a chunk, and let $M_0$ and $I_0$ be the final states from the previous chunk. The states are computed via the associative operator $\oplus$:

$$(M_b, \bar{I}_b, A_b) \oplus (M_a, \bar{I}_a, A_a) = (M_b + A_b M_a, \bar{I}_b + A_b \bar{I}_a, A_a A_b) \tag{35}$$

The posterior mean is recovered as $\mu_t = M_t / I_t$, followed by an *einsum* with the query $q_t$ to produce the output.

A primary limitation of Palimpsa is the necessity to explicitly store all Intra-Chunk states $M_t$ and $I_t$ in registers to perform the output computation. This requirement imposes a heavy register memory footprint, creating a significant bottleneck. To mitigate this register pressure, we are constrained to use smaller chunk dimensions $B_T$, which negatively impacts training throughput.

Furthermore, all Inter-Chunk states must be materialized in HBM, which can restrict the maximum batch size during training as the state size and sequence length scale. This creates a fundamental trade-off: a larger $B_T$ reduces HBM traffic but significantly increases register pressure.

In practice, the performance overhead is size-dependent: for a 170M parameter model, the training time is approximately $1.5\times$ that of an optimized Simple GLA from the FLA (Yang et al., 2024) kernel, while for a 760M model, this ratio increases to approximately $3\times$.

### A.13. Inference Efficiency

While Palimpsa's training throughput is constrained by register pressure from the parallel associative scan, these bottlenecks vanish at inference time. During generation, the model operates in its native recurrent mode, processing tokens sequentially. This eliminates the need to materialize intermediate states across large chunks, reducing the memory footprint to a constant state size per step.

We benchmarked the fused recurrent kernel on a single **NVIDIA GeForce RTX 3090** GPU ($B = 16, L = 2048, H = 1$), varying the model dimension $D \in \{512, 1024, 2048\}$. As shown in Figure 7, Palimpsa demonstrates robust throughput, peaking at roughly 770 kt/s for $D = 512$ and maintaining a consistent factor of $\approx 4\times$ slowdown compared to Simple GLA as the dimension scales to 2048 ($\sim 50$ kt/s vs $\sim 188$ kt/s).

It is important to note that this benchmark isolates the attention mechanism. In the context of a full model inference pass, the relative performance gap would narrow significantly, as a substantial portion of the compute budget is consumed by the Gated MLP, embeddings, and layer normalization, which are identical across both architectures.

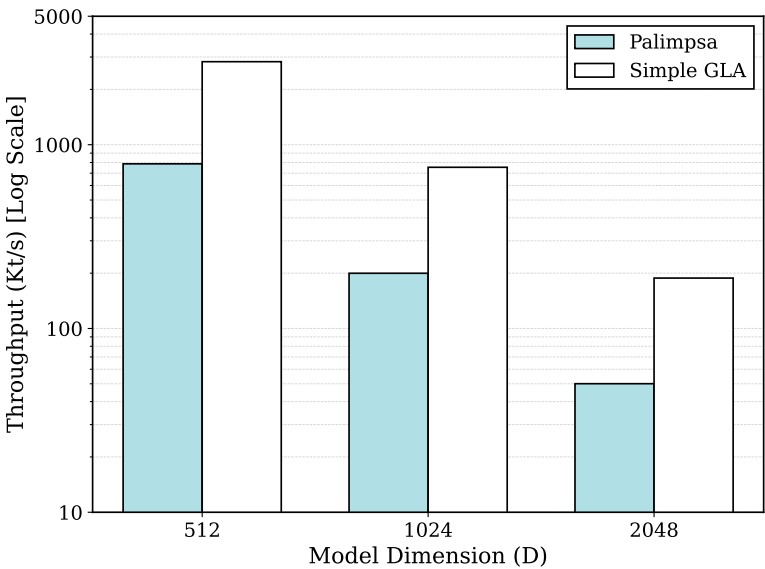

*Figure 7.* **Inference Speed Benchmark.** Throughput (thousands of tokens/s) comparison between Palimpsa and Simple GLA on an NVIDIA GeForce RTX 3090. Palimpsa matches the baseline's scaling behavior while maintaining a consistent $4\times$ factor due to the dual-state update overhead.

