# OpenReview forum: "Learning to Remember, Learn, and Forget in Attention-Based Models"
_ICML.cc/2026/Conference — ICML 2026 regular_

### Official Review · Reviewer_pT6J · 2026-03-06

**Soundness:** 3
**Presentation:** 3
**Significance:** 3
**Originality:** 3
**Overall Recommendation:** 5
**Confidence:** 4

**Summary:**

The paper addresses the core issues of Gated Linear Attention models in Transformers, namely their susceptibility to interference due to fixed memory capacity and the problem of catastrophic forgetting/remembering in long sequence processing. By treating In-Context Learning (ICL) as a continual learning problem and introducing the neuro-inspired concept of meta-plasticity, the authors propose Palimpsa, a self-attention model based on Bayesian meta-plasticity. This model performs Bayesian updates of meta-plasticity through dual-state attention blocks, dynamically adjusting the plasticity of memory states and releasing obsolete knowledge. Theoretically, it proves that Mamba2 is a special case of Palimpsa dominated by forgetting, and it constructs a unified Bayesian mathematical framework that incorporates existing gated linear attention models, enabling the transformation of non-meta-plasticity models into meta-plasticity ones. In experiments, two variants—Palimpsa-D (based on Deltanet) and Palimpsa-M (based on Mamba2)—consistently outperform baseline models like Gated Deltanet on associative recall tasks in the MQAR benchmark and on common sense reasoning tasks after pre-training on Fineweb-Edu. At the 760M parameter scale, Palimpsa-D achieves an average accuracy of 52.27%, surpassing the baseline by nearly 0.8 percentage points. Furthermore, the model's localized update rules are compatible with efficient hardware architectures, providing a memory-efficient solution for sequence modeling on edge devices.

**Compliance With Llm Reviewing Policy:**

Affirmed.

**Key Questions For Authors:**

1. The paper reports that Palimpsa’s training throughput is 3× slower than Mamba2 for 760M models due to register pressure from parallel associative scans. Have you explored optimization strategies to mitigate this training overhead (e.g., adaptive chunk size tuning, hardware-specific kernel optimizations, or relaxed approximations of the precision matrix)? How much improvement in training speed could these strategies achieve without sacrificing metaplasticity benefits?

2. You demonstrate Mamba2 is a special case of Palimpsa where forgetting dominates. Have you quantified how the forgetting rate (controlled by Nt​) impacts performance across different task types (e.g., short-vs-long sequence tasks, fact recall vs. commonsense reasoning)? Is there an optimal range of forgetting rates that generalizes across tasks, or does it require task-specific tuning?

**Strengths And Weaknesses:**

**Strengths**

1. **Theoretical Rigor:** The paper’s core contribution—framing In-Context Learning (ICL) as a continual learning problem via Bayesian metaplasticity—is technically sound. The derivation of Palimpsa’s update rules, variational free energy decomposition, and diagonal approximation of covariance matrices is mathematically rigorous, with detailed appendices (A.2–A.7) validating key steps (e.g., Bayesian forgetting, closed-form solutions for mean and precision matrices). Assumptions (e.g., Gaussian distributions for attention states) are standard in the field and well-justified.

2. **Addressing a Critical, Practical Problem:** The paper targets a well-recognized bottleneck of fixed-size attention models (e.g., linear transformers, Mamba) that limits their utility in long-context and edge applications. By solving catastrophic forgetting/remembering, Palimpsa unlocks these models for use cases like edge AI, long-document processing, and robotics (where memory efficiency is non-negotiable).

3. **Dual Impact: Theory & Practice:** Theoretically, it unifies disjoint gated linear attention/SSM models under a single Bayesian framework, revealing Mamba2 as a special case—this deepens the field’s understanding of existing methods. Practically, the “pre-train (non-metaplastic) + fine-tune (metaplastic)” pipeline allows seamless upgrading of existing models (e.g., Mamba2) without re-training, offering immediate value to practitioners.

**Weaknesses**

1. **Limited Visual Intuition:** While Figure 1 illustrates the Palimpsa layer, the paper lacks visualizations of how metaplasticity dynamically adjusts memory states (e.g., before/after forgetting gate activation) to reinforce key insights.

2. **Modest Performance Gains in Some Settings:** For 170M models, the average accuracy improvement over baselines is modest (~0.5 points). The paper could better emphasize that gains grow with sequence length and model scale (760M models see ~0.8 points over Gated Deltanet) to highlight significance.

3.

---

> ### Author Rebuttal · Authors · 2026-03-30
>
> We thank Reviewer 3 for the appreciation of our theoretical rigor and for recognizing the dual theoretical and practical impact of Palimpsa. We address your questions below:
>
> * **Modest Performance Gain:** Thank you for pointing out that indeed the gain seems to be growing with the scale. This trend continues with the 2.7B model we recently trained (See R1.3 and Table 1). Looking at the average accuracy gap between Mamba2 (without metaplasticity) and our fine-tuned Palimpsa-M, the gain grows from +0.58 points at 170M, to +0.63 points at 760M, and reaches +0.84 points for our newly trained 2.7B model. We will ensure this tendency is better emphasized in the revised text.
>
> * **Visual Intuition and Impact of Forgetting Rate:** We agree that more visual intuition would strengthen the paper. In the revision, we include a heatmap illustrating the distribution of the memory window $N_t$ across different layers and sequence steps for the 2.7B language model we recently trained, providing a clear visual intuition of how the model layers dynamically forget information. We did not try to constrain $N_t$ to evaluate its impact on various tasks. The range was therefore $[1, \infty)$. We expect that constraining $N_t$ to stay relatively small would indeed impede the model's ability on long-context tasks, and forcing $N_t$ to be relatively large would weaken short-term dependencies. Our experiments revealed that, during training, the language model naturally learns an $N_t$ range spanning from close to the minimum up to approximately 600K (See [Figure](https://pasteboard.co/7t2vgyj96UzC.png)).
>
> * **Training Overhead and Optimization Strategies:** This is a critical point. So far, we have only used Triton autotuning to optimize efficiency; we have identified several promising strategies to mitigate the training overhead. Our latest experiments (see answer to previous reviewer) addressed this problem architecturally on a 2.7B parameter model. We selectively deployed Palimpsa layers in the top-8 highest-$N$ layers where metaplasticity is most beneficial, using standard fast kernels for the rest. Algorithmically, we could apply a relaxed approximation: by assuming the importance matrix remains constant within a small chunk, we could theoretically halve the register pressure. We will add a dedicated discussion of these optimization pathways to the manuscript.
>
>
> **Table 1:** Performances of Palimpsa-M and Mamba2 on language modeling and Commonsense Reasoning tasks. The best results are in bold font. ppl: perplexity, acc: accuracy, acc_n: normalized accuracy. All models are 2.7B variants.
>
> | Model | Wiki. ppl ↓ | LMB. ppl ↓ | LMB. acc ↑ | PIQA acc_n ↑ | Hella. acc_n ↑ | Wino. acc ↑ | ARC-e acc ↑ | ARC-c acc_n ↑ | SIQA acc ↑ | Avg. acc ↑ |
> | :--- | :---: | :---: | :---: | :---: | :---: | :---: | :---: | :---: | :---: | :---: |
> | Mamba2 | **14.46** | **4.79** | 62.82 | 75.73 | 68.12 | 63.85 | 60.06 | 39.16 | **44.78** | 59.22 |
> | Palimpsa-M | 14.51 | **4.79** | **63.26** | **76.17** | **68.58** | **65.19** | **63.43** | **39.33** | 44.47 | **60.06** |
>
> **Table 2:** Performances of Palimpsa-M and Mamba2 on LongBench trained on 315B tokens (instruction tuned with LongAlpaca) with 2.7B parameters. The best results are in bold.
>
> | Model | easy | hard | short | medium | long | overall |
> | :--- | :---: | :---: | :---: | :---: | :---: | :---: |
> | Mamba2 | **22.4** | 17.4 | **18.9** | 19.5 | 19.4 | 19.3 |
> | Palimpsa-M | **22.4** | **18.6** | 16.1 | **23.3** | **20.4** | **20.4** |
>
> **Table 3:** RULER Benchmark of Palimpsa-M and Mamba2 grouped by task category: Retrieval (N-1 to MK-N), Tracing (VT), Aggregation (CWE, FWE), and QA (HotPot, SQA). The best results are in bold.
>
> | Model | N-1 | N-2 | N-3 | MQ-N | MV-N | MK-N | VT | CWE | FWE | HotPot | SQA |
> | :--- | :---: | :---: | :---: | :---: | :---: | :---: | :---: | :---: | :---: | :---: | :---: |
> | **4k Context** | | | | | | | | | | | |
> | Mamba2 | **100.0** | **88.0** | **83.6** | 53.1 | 51.5 | 25.0 | **59.6** | **30.5** | **64.0** | 25.8 | **28.8** |
> | Palimpsa-M | **100.0** | 84.4 | 77.2 | **57.5** | **59.9** | **28.4** | 33.5 | 29.6 | 58.4 | **29.8** | 28.4 |
> | **8k Context** | | | | | | | | | | | |
> | Mamba2 | **100.0** | **46.2** | **42.4** | 38.8 | 34.9 | **24.6** | **49.7** | **20.5** | **64.0** | **24.4** | 19.0 |
> | Palimpsa-M | **100.0** | 44.8 | **42.4** | **39.4** | **37.2** | 21.8 | 24.9 | 13.0 | 56.7 | 23.8 | **19.2** |
> | **16k Context** | | | | | | | | | | | |
> | Mamba2 | **100.0** | 9.4 | **13.6** | 7.0 | 6.0 | **8.0** | **26.0** | **2.4** | **59.6** | **18.4** | 15.6 |
> | Palimpsa-M | 99.4 | **11.2** | 12.6 | **8.2** | **9.7** | **8.0** | 22.5 | 0.6 | 43.8 | 17.0 | **17.1** |
> | **32k Context** | | | | | | | | | | | |
> | Mamba2 | **94.4** | 7.0 | **9.4** | 2.3 | 1.1 | 7.2 | 14.9 | **0.4** | **31.9** | 20.2 | 14.7 |
> | Palimpsa-M | 93.0 | **7.6** | 8.2 | **2.8** | **1.4** | **7.6** | **24.4** | 0.2 | 24.7 | **20.6** | **15.1** |

---

### Official Review · Reviewer_PUK1 · 2026-03-12

**Soundness:** 3
**Presentation:** 3
**Significance:** 3
**Originality:** 3
**Overall Recommendation:** 4
**Confidence:** 3

**Summary:**

Summary of "Learning to Remember, Learn, and Forget in Attention-Based Models"
This paper introduces Palimpsa, an attention block for sequence models that frames In-Context Learning (ICL) as a continual learning problem. The authors observe that gated linear attention models with fixed-size memory states are subject to catastrophic forgetting and catastrophic remembering — failure modes in which new information overwrites old associations, or accumulation of prior weight prevents integration of new information. Palimpsa addresses this tension via Bayesian metaplasticity, where each memory state is paired with an importance state derived from a variational posterior, and the learning rate of each state adapts to its uncertainty.
The derivation proceeds from a variational free energy objective over a weighted posterior that incorporates a forgetting mechanism. The authors show that the resulting update equations decompose into three terms: a plasticity term for new information, a forgetting term controlled by a window parameter, and a stability term that penalizes deviation from prior states in proportion to their precision. By applying a diagonal approximation to the precision matrix, the authors obtain update rules that are computable in closed form and parallelizable via associative scan. The paper further demonstrates that Mamba2, Deltanet, Gated Deltanet, Longhorn, and MesaNet each emerge as special cases or approximations within this framework.
Experiments are conducted on the Multi-Query Associative Recall benchmark and on language modeling at 170M and 760M parameter scales using the FineWeb-Edu dataset. On MQAR, Palimpsa variants outperform non-metaplastic counterparts, with the performance gap growing as sequence length increases. On Commonsense Reasoning tasks, Palimpsa-D at 760M achieves 52.27% average accuracy, surpassing the Gated Deltanet baseline. The authors also propose a fine-tuning pipeline that converts a pre-trained Mamba2 model into a metaplastic one, exploiting a continuum between the two architectures to reduce training cost.

**Compliance With Llm Reviewing Policy:**

Affirmed.

**Final Justification:**

The authors have adequately addressed the major concerns. The addition of LongBench and RULER evaluations directly validates the paper's core motivation, with Palimpsa showing over 10-point gains on variable tracking at 32k context. Theoretical contributions remain solid, and the fine-tuning pipeline has clear practical value.
Remaining concerns — the unverified hypothesis for the fully-trained vs. fine-tuned gap, lack of Iprior sensitivity analysis, and absent direct comparison with Longhorn — are limitations of scope rather than fundamental flaws.
I maintain 4: Weak Accept, provided the camera-ready version includes: explicit discussion of the fully-trained vs. fine-tuned reversal as a limitation, metaplasticity ratio defined before Figure 3, bscale fully described in the main text, and the LongBench/RULER results incorporated into the main body.

**Key Questions For Authors:**

1. The fully trained Palimpsa-D underperforms the fine-tuned variant on several benchmarks at both the 170M and 760M scales. What is the explanation for this?

2. The core motivation of the paper is improved memory retention over long sequences, but no long-context evaluation is included. Tasks such as SCROLLS, RULER, or a simple needle-in-a-haystack retrieval test would directly measure whether the metaplastic memory window translates to better performance on long inputs. Would adding such an evaluation be feasible?

3. The inference throughput benchmark is run at batch size 1 and sequence length 32. At these settings, memory bandwidth is not a bottleneck and the result does not reflect the conditions under which edge deployment would occur. Can you provide throughput numbers at batch sizes and sequence lengths that are more representative of inference in practice, for example batch size 8 or 16 and sequence lengths of 512 or 2048?

**Limitations:**

No.
The authors discuss computational limitations honestly — the register pressure from the parallel associative scan, the training throughput overhead relative to Mamba2, and the HBM bottleneck at large batch sizes are all acknowledged. This is a positive aspect of the paper.
However, the limitations section does not address several issues that are relevant to the work. The paper does not discuss the sensitivity of the method to the choice of Iprior, which controls the forgetting horizon and is likely to require careful tuning across different tasks and sequence length regimes. There is no discussion of failure cases — specifically, under what conditions metaplasticity does not help or actively hurts performance. The gap between fully trained and fine-tuned variants is not mentioned as a limitation despite being visible in Table 2. The restriction of the evaluation to a narrow set of benchmarks, and the absence of long-context tasks, is not acknowledged as a limitation of the empirical claims.

**Strengths And Weaknesses:**

This paper reframes In-Context Learning as a continual learning problem and addresses the stability-plasticity dilemma in fixed-size memory models through Bayesian metaplasticity. The core contributions are three: deriving the Palimpsa update rule in closed form from a variational free energy objective, unifying several existing architectures including Mamba2 as special cases within a single Bayesian framework, and proposing a fine-tuning pipeline that converts a pre-trained Mamba2 model into a metaplastic one. The derivation is given in full in the appendix, and the claim that Mamba2 is a special case of Palimpsa is supported algebraically rather than asserted. Experiments are run over 8 random seeds with standard deviations reported, and the contribution of metaplasticity is isolated via ablation with all other factors held fixed. The authors' candid acknowledgment of training throughput costs is also a positive sign.
There are, however, several limitations. The MQAR experiments use a two-layer model with hidden dimension 128, and it is not clear whether gains at this scale carry over to full language models. Performance improvements in language modeling are small in absolute terms — 0.6 to 0.8 points — and there are cases where fully trained Palimpsa-D underperforms its fine-tuned counterpart, which the paper does not analyze. The inference throughput benchmark is conducted on a single GPU with batch size 1 and sequence length 32, which does not reflect realistic deployment conditions. MesaNet is excluded from comparisons due to reported numerical instabilities in the public code, which omits the closest competing model. On presentation, the metaplasticity ratio appears in Figure 3 before it is defined in the main text, and the description of the bscale parameter central to the fine-tuning procedure is split between the main text and the appendix in a way that hinders reproducibility. On originality, the diagonal precision update closely follows MESU and Longhorn, and the distinction from Longhorn in particular is not sufficiently supported by direct experimental comparison. Finally, there is no evaluation on long-context tasks that would directly measure the memory improvement that motivates the work.
Overall, the paper makes a technically sound and conceptually clear contribution. The unifying framework and the fine-tuning pipeline in particular have practical value. Strengthening the experimental scale, completing the model comparisons, and aligning the experimental design more directly with the core motivation would substantially improve the paper's persuasiveness.

---

> ### Author Rebuttal · Authors · 2026-03-30
>
> Thank you for your encouraging review and constructive feedback. We address your specific concerns below:
>
> * **Fully-Trained vs. Fine-Tuned Palimpsa:** Indeed, fine-tuned variants sometimes outperform the fully-trained ones. Our hypothesis is that during early stages of pre-training, models primarily learn basic token marginal probabilities and short-term dependencies, i.e. a regime where the complex dynamics of metaplasticity are unnecessary and potentially harder to optimize from a random initialization. Starting with a simpler update rule (Mamba2), followed by Palimpsa when longer-range dependencies are required, might act as an effective curriculum. We will explicitly add this hypothesis and note it as a limitation in the revised text.
>
> * **Long-Context Evaluation (LongBench, RULER):** We agree that a long-context evaluation was missing. We now show results on a fine-tuned 2.7B pre-trained Mamba2 model, evaluated on the LongBench benchmark and the RULER benchmark. On LongBench, Palimpsa achieves a +1.1 point overall improvement, including a +3.8 point gain in the Medium category (32k - 128k words). This confirms that the metaplastic memory window directly translates to better long-context retention (see Table 2 R3, and R1.3 for fine-tuning details).
>
>   On the suggested RULER benchmark (see Table 3 R3), the results are more mixed. In the Retrieval category, a clear trend emerges: while Mamba2 surpasses Palimpsa in single retrieval, Palimpsa consistently wins at MQ-N: retrieving all needles with distinct keys, MV-N: retrieving all values sharing the same key, and MK-N: retrieving one target needle while ignoring hard distractor needles. We speculate this is because Mamba2's strong selectivity effectively protects a single vital piece of information, while Palimpsa's metaplasticity provides a better compromise between remembering and learning when multiple pieces of information in the context are important.
>
>   RULER contains a set of benchmark tasks that aim to test different aspects of long-context performance of language models. For fixed-size memory models like linear attention and SSMs, variable tracking (VT) represents a particularly representative and difficult task. It directly measures the "memory-management" aspect of the fixed-size memory which is the goal of Palimpsa.
>
>   These results support our claim that Palimpsa memory management is more resilient in the long-context regime. While it lacks performance compared to Mamba2 at small 4k context, it surpasses Mamba2 by more than 10 points when the context approaches 32k tokens. On aggregation tasks (e.g. where the model must extract the most frequent words from the context), Mamba2 is better. It appears that the introduction of metaplasticity hurts this specific capacity, though the exact reason remains an open question. Finally, for Question Answering, the results are mixed.
>
> * **Inference Throughput Benchmark:** We agree that these settings are more representative of inference in practice. We performed the requested experiment with Batch-size=16 and sequence Length=2048, and updated the supplementary note accordingly. The results remain unchanged; the approximate 4-fold ratio is maintained (see [Figure](https://pasteboard.co/mdUAAUrXZ1N9.png)). At the highest dimension, Palimpsa stayed at 50Kt/s while SimpleGLA slightly dropped to 188Kt/s.
>
> * **Presentation and Baseline Comparisons:** We will fix the text so the metaplasticity ratio is defined before Figure 3, and ensure that $b_{scale}$ is fully defined in the main text. Regarding baseline comparisons, we chose not to benchmark against Longhorn because our Gated DeltaNet (GDN) baseline strictly outperforms Longhorn. Due to computational limitations, we opted to evaluate Palimpsa against the strongest available non-metaplastic baseline. As for MESU, this algorithm was not developed for linear attention but for training Bayesian neural network parameters and relies on Monte-Carlo sampling and backpropagation, which would be intractable without modification.
>
> * **Limitations ($I_{prior}$ and Failure Cases):** Thank you for pointing out these missing discussions. In our experiments, $I_{prior}$ was fixed to 1. During our early explorations of Palimpsa, we found that making it learnable did not correlate with any improvements. We will mention this in our main text in the revised manuscript.

---

### Official Review · Reviewer_KSvu · 2026-03-12

**Soundness:** 3
**Presentation:** 2
**Significance:** 2
**Originality:** 2
**Overall Recommendation:** 5
**Confidence:** 3

**Summary:**

The work explores in-context learning (ICL) with linear-attention models from a continual learning perspective.
The challenge for this models is to decide how to update the memory between sequence inputs at test-time, and how to mitigate catastrophic forgetting.
The authors propose to look at the attention component via a Bayesian formulation (similarly to Bonnet et al.). Assuming Gaussian distributions, they derive a weighted posterior via variational inference. This leads to their layer that requires to track the mean and a precision (covariance) diagonal matrix. These are updated recursively each timestep. Effectively, this method introduces a forgetting gate. This adds the idea they refer to as metaplasticity which mitigates catastrophic forgetting and remembering. The method is dubbed “Palimpsa”.  Additionally, the authors connect such framework to previous works to show the extendability of the definition of the framework.

Last, the method is tested in synthetic experiments using the Multi-Query Associate Recall benchmark, showing a slight improvement over Yang et al (Gated Deltanets). They explore the dynamics of palimpsa’s metaplasticity during training. Last the authors, test and show how to translate existing related models by  fine-tuning towards Palimpsa. They show small improvements after fine-tuning on language modeling and commonsense reasoning tasks.

**Compliance With Llm Reviewing Policy:**

Affirmed.

**Final Justification:**

The rebuttal addressed my main concerns, in particular with respect to long context evaluation.
I would appreciate the authors' intentions to improve the presentation for the final version.

**Key Questions For Authors:**

* How does your method performs on long-context tasks, and compared to non-linear attention models?
* Is there a way we could measure the metaplasticity activity in the network to compare how each model (previous work and palimpsa) do?

**Limitations:**

yes

**Strengths And Weaknesses:**

Strengths:
* The work provides a refreshing perspective to the family of linear attention models through the Bayesian formulation.
* The presentation is well written and clear, in general.
* The solution addresses a potential problem when using linear attention models for (long) sequential modeling.
* Palimpsa incrementally builds on top of previous work, overcoming its limitations.

Weaknesses:
* The flow of the text could be greatly improved. Continuously, there is a back and forth between Palimpsa and other works, interfering with the derivation of the method. Also, it would be good to summarize the Palimpsa layer equations in a single place.
* It is unclear how much representational power is lost by using a diagonal precision matrix to simplify the updates.
* The results in the experimental section show small improvements. There is no discussion on what are the reasons to achieve better results. Additionally, it would be interesting to focus on benchmarks that require long context.

---

> ### Author Rebuttal · Authors · 2026-03-30
>
> Thank you for your encouraging review and constructive feedback.
>
> * **Text Flow and Equation Summary:** We agree that the manuscript structure can be improved, and will implement the requested changes. In particular, we will summarize the Palimpsa equations in one place and condense specific parts of the text into dedicated sections to maintain flow.
>
> * **Representational Power:** The diagonal approximation (which effectively treats synapses as independent) indeed impacts representational power, particularly for recall ability. We observe this empirically on the MQAR benchmark: Palimpsa-D *without* metaplasticity, which can be seen as Gated DeltaNet (GDN) with the diagonal approximation, performs worse than the GDN baseline. However, Palimpsa-D *with* metaplasticity outperforms GDN, demonstrating that the metaplasticity effectively compensates for and overcomes this representational bottleneck.
>
>   To further improve results, a promising future direction is treating GDN and Palimpsa as complementary (e.g., using GDN for low-$N$ layers and Palimpsa for high-$N$ layers). In the following paragraph, we show a related direction for Palimpsa-M.
>   Alternatively, we could explore a mean-field approximation, thereby maintaining a diagonal covariance matrix while retaining the delta rule projection ($\mu_{t-1}k_t$) for the update on an additional "slow state" that is updated only every $K$ steps. This would help recover representational capacity while keeping computational costs manageable.
>
> * **Performance on Long-Context Tasks and Scale:** We agree that the lack of long context evaluation is a weakness of our manuscript. To correct this, and given the short time frame and vast resource requirements we opted for a continued pre-training and model surgery experiment, starting from the 2.7B Mamba2 checkpoint trained on 300B tokens.
>   Since Mamba2 is a limiting case of Palimpsa, we can compute the hypothetical $N_t$ for every layer, i.e. the strength of forgetting.
>   Palimpsa should have the strongest impact on the layers with the largest $N_t$, i.e. weakest forgetting.
>   We leveraged this hypothesis to identify the top-8 layers in the 64-layer stack of Mamba2-2.7B and replace them with Palimpsa, preserving common parameters and designating them as trainable.
>   We continued training for an additional 10B tokens on a mixture of FineWebEdu and StackV2.
>   Next, we moved on to long-context extension with 32k tokens by using a mixture of 5B tokens of long-form text from the "arxiv" and "book" subsets of SlimPajamas, repository-level concatenations of StackV2 C++ and Python code, and Needle-in-a-Haystack data to adapt the state to the new time horizon [1, 2].
>   Finally, the model was instruction fine-tuned on LongAlpaca [3].
>   We then evaluated the model on common language benchmarks, RULER, and LongBench to demonstrate the requested long-range performance.
>   We concurrently applied the same training recipe to Mamba2 to guarantee a fair comparison.
>   The details of this long-context pre-training procedure will be in the revised manuscript.
>
>   Results show that Palimpsa improves over Mamba2 on commonsense reasoning (+0.84 points), (see Table 1 in R3). While a 0.84 point improvement might look small, we can put it in perspective. Scaling Mamba2 by 3.6x in parameters and 10x in training data only gives a 7.7 point improvement. This means gaining 0.84 points just from the architecture alone actually saves significant storage and compute. Palimpsa also shows significant gains on the long-context LongBench benchmark (+1.1 points overall). Notably, Palimpsa outperforms the baseline by nearly 4 points in the Medium category (see Table 2 in R3), corresponding to context lengths between 32k and 128k words. We note that a non-linear model (e.g. standard softmax attention) performs better on this benchmark; however, at a high computational and memory cost at such sequence lengths. As an example, Llama 3.1B reports an overall accuracy of about 30%.
>
> * **Measuring Metaplasticity Activity:** Metaplasticity is the concept of plasticity being itself plastic; with this understanding, previous architectures (like Gated DeltaNets or Longhorn) cannot exhibit metaplastic behavior because the learning rate of synapses is not learned. For Palimpsa, as detailed in Section 3.2 (Figure 3), we track the metaplasticity ratio, which is defined as $(I_{max}-I_{min})/I_{min}$ to quantify how the difference between the most plastic and the most consolidated synapses varies over time. A Palimpsa layer with a metaplasticity ratio of 0 is perfectly equivalent to Mamba2. We clarified this in the revised manuscript.
>
> **References:**
> [1] How to Train Long-Context Language Models (Effectively), Gao et al., 2025, ACL
> [2] Enabling High-Sparsity Foundational Llama Models with Efficient Pretraining and Deployment, Agarwalla et al., 2024., arxiv
> [3] LongLoRA: Efficient Fine-tuning of Long-Context Large Language Models, Chen et al., 2023, ICLR

---

> > ### Author Rebuttal · Reviewer_KSvu · 2026-04-02
> >
> > My score has been revisited

---

### Decision · Program_Chairs · 2026-04-30

**Decision:**

Accept (regular)

**Comment:**

Palimpsa frames in-context learning as a continual learning problem and proposes a Bayesian metaplasticity update for gated linear attention. I think the theoretical framing is nice, and it is a strength that Mamba2, DeltaNet, Gated DeltaNet, and related models appear as special cases within a unified framework. The fine-tuning pipeline that converts a pre-trained Mamba2 model into a metaplastic one is also a practical contribution. All three reviewers recommend acceptance (5, 4, 5), and the rebuttal addressed most of their concerns. In particular, the authors added larger-scale experiments and long-context evaluations on LongBench and RULER, which were important missing pieces. The empirical gains are generally modest but consistent, and there is some evidence that they improve with scale. Overall, I think this is a clear accept. For the camera-ready, I would ask the authors to follow up on reviewer PUK1’s suggestions: move the long-context results into the main body, define the metaplasticity ratio before Figure 3, describe bscale fully in the main text, and note the fully-trained vs. fine-tuned gap as a limitation.